# Order-Invariant Cardinality Estimators Are Differentially Private

**Charlie Dickens**
Yahoo
charlie.dickens@yahooinc.com

**Justin Thaler**
Georgetown University
justin.thaler@georgetown.edu

**Daniel Ting**
Meta*
dting@meta.com

## Abstract

We consider privacy in the context of streaming algorithms for cardinality estimation. We show that a large class of algorithms all satisfy $\epsilon$-differential privacy, so long as (a) the algorithm is combined with a simple down-sampling procedure, and (b) the input stream cardinality is $\Omega(k/\epsilon)$. Here, $k$ is a certain parameter of the sketch that is always at most the sketch size in bits, but is typically much smaller. We also show that, even with no modification, algorithms in our class satisfy $(\epsilon, \delta)$-differential privacy, where $\delta$ falls exponentially with the stream cardinality. Our analysis applies to essentially all popular cardinality estimation algorithms, and substantially generalizes and tightens privacy bounds from earlier works. Our approach is faster and exhibits a better utility-space tradeoff than prior art.

## 1 Introduction

Cardinality estimation, or the distinct counting problem, is a fundamental data analysis task. Typical applications are found in network traffic monitoring [9], query optimization [20], and counting unique search engine queries [14]. A key challenge is to perform this estimation in small space while processing each data item quickly. Typical approaches for solving this problem at scale involve data sketches such as the Flajolet-Martin (FM85) sketch [12], HyperLogLog (HLL) [11], Bottom-$k$ [2, 6, 3]. All these provide approximate cardinality estimates but use bounded space.

While research has historically focused on the accuracy, speed, and space usage of these sketches, recent work examines their privacy guarantees. These privacy-preserving properties have grown in importance as companies have built tools that can grant an appropriate level of privacy to different people and scenarios. The tools aid in satisfying users' demand for better data stewardship, while also ensuring compliance with regulatory requirements.

We show that *all* cardinality estimators in a class of hash-based, order-invariant sketches with bounded size are $\epsilon$-differentially private (DP) so long as the algorithm is combined with a simple down-sampling procedure and the true cardinality satisfies a mild lower bound. This lower bound requirement can be guaranteed to hold by inserting sufficiently many "phantom elements" into the stream when initializing the sketch. We also show that, even with no modification, algorithms in our class satisfy $(\epsilon, \delta)$-differential privacy, where $\delta$ falls exponentially with the stream cardinality.

Our novel analysis has significant benefits. First, prior works on differentially private cardinality estimation have analyzed only specific sketches [23, 25, 5, 22]. Moreover, many of the sketches analyzed (e.g., [23, 22]), while reminiscent of sketches used in practice, in fact differ from practical

36th Conference on Neural Information Processing Systems (NeurIPS 2022).

sketches in important ways. For example, Smith et al. [22] analyze a *variant* of HLL that Section 4 shows has an update time that can be $k$ times slower than an HLL sketch with $k$ buckets.

While our analysis covers an entire class of sketches at once, our error analysis improves upon prior work in many cases when specialized to specific sketches. For example, our analysis yields tighter privacy bounds for HLL than the one given in [5], yielding both an $\epsilon$-DP guarantee, rather than an $(\epsilon, \delta)$-DP guarantee, as well as tighter bounds on the failure probability $\delta$—see Section 4 for details. Crucially, the class of sketches we analyze captures many (in fact, almost all to our knowledge) of the sketches that are actually used in practice. This means that existing systems can be used in contexts requiring privacy, either without modification if streams are guaranteed to satisfy the mild cardinality lower bound we require, or with a simple pre-processing step described if such cardinality lower bounds may not hold. Thus, existing data infrastructure can be easily modified to provide DP guarantees, and in fact existing sketches can be easily migrated to DP summaries.

## 1.1 Related work

One perspective is that cardinality estimators cannot simultaneously preserve privacy and offer good utility [7]. However, this impossibility result applies only when an adversary However, this impossibility result applies only when an adversary can create and merge an arbitrary number of sketches, effectively observing an item's value many times. It does not address the privacy of one sketch itself.

Other works have studied more realistic models where either the hashes are public, but private noise is added to the sketch [23, 17, 25], or the hashes are secret [5] (i.e., not known to the adversary who is trying to "break" privacy). This latter setting turns out to permit less noisy cardinality estimates. Past works study specific sketches or a variant of a sketch. For example, Smith et al. [22] show that an HLL-type sketch is $\epsilon$-DP while [25] modifies the FM85 sketch using coordinated sampling, which is also based on a private hash. Variants of both models are analyzed by Choi et al. [5], and they show (amongst other contributions) a similar result to [22], establishing that an FM85-type sketch is differentially private. Like these prior works, we focus on the setting when *the hash functions are kept secret* from the adversary. A related problem of differentially private estimation of cardinalities under set operations is studied by [18], but they assume the inputs to each sketch are already de-duplicated.

There is one standard caveat: following prior works [22, 5] our privacy analysis assumes a perfectly random hash function. One can remove this assumption both in theory and practice by using a cryptographic hash function. This will yield a sketch that satisfies either a computational variant of differential privacy called SIM-CDP, or standard information-theoretic notions of differential privacy under the assumption that the hash function fools space-bounded computations [22, Section 2.3].

Other works also consider the privacy-preserving properties of common $L_p$ functions over data streams. For $p = 2$, these include fast dimensionality reduction [4, 24] and least squares regression [21]. Meanwhile, for $0 < p \leq 1$, frequency-moment estimation has also been studied [26]. Our focus is solely the cardinality estimation problem when $p = 0$.

## 1.2 Preliminaries

More formally, we consider the following problem.

**Problem Definition**  Let $\mathcal{D} = \{x_1, \ldots, x_n\}$ denote a stream of samples with each identifier $x_i$ coming from a large universe $U$, e.g., of size $2^{64}$. The objective is to estimate the cardinality, or number of distinct identifiers, of $\mathcal{D}$ using an algorithm $S$ which is given privacy parameters $\epsilon, \delta \geq 0$ and a space bound $b$, measured in bits.

**Definition 1.1** (Differential Privacy [8])**.** *A randomized algorithm $S$ is $(\epsilon, \delta)$-differentially private ($(\epsilon, \delta)$-DP for short or if $\delta = 0$, pure $\epsilon$-DP) if for any pair of data sets $\mathcal{D}, \mathcal{D}'$ that differ in one record and for all $S$ in the range of $S$, $\Pr(S(\mathcal{D}') \in S) \leq e^\epsilon \Pr(S(\mathcal{D}) \in S) + \delta$ with probability over the internal randomness of the algorithm $S$.*

Rather than analyzing any specific sketching algorithm, we analyze a natural class of randomized distinct counting sketches. Algorithms in this class operate in the following manner: each time a new stream item $i$ arrives, $i$ is hashed using some uniform random hash function $h$, and then $h(i)$ is used to update the sketch, i.e., the update procedure depends only on $h(i)$, and is otherwise independent of

$i$. Our analysis applies to any such algorithm that depends only on the *set* of observed hash values. Equivalently, the sketch state is invariant both to the order in which stream items arrive, and to item duplication.[2] We call this class of algorithms *hash-based, order-invariant* cardinality estimators. Note that for any hash-based, order-invariant cardinality estimator, the distribution of the sketch depends only on the cardinality of the stream. All distinct counting sketches of which we are aware that are invariant to permutations of the input data are included in this class. This includes FM85, LPCA, Bottom-$k$, Adaptive Sampling, and HLL as shown in Section 4.

**Definition 1.2** (Hash-Based, Order-invariant Cardinality Estimators)**.** *Any sketching algorithm that depends only on the* set *of hash values of stream items using a uniform random hash function is a* hash-based order-invariant cardinality estimator. *We denote this class of algorithms by* $\mathcal{C}$.

We denote a sketching algorithm with internal randomness $r$ by $S_r$ (for hash-based algorithms, $r$ specifies the random hash function used). The algorithm takes a data set $\mathcal{D}$ and generates a data structure $S_r(\mathcal{D})$ that is used to estimate the cardinality. We refer to this structure as the *state of the sketch*, or simply the *sketch*, and the values it can take by $s \in \Omega$. Sketches are first initialized and then items are inserted into the sketch with an `add` operation that may or may not change the sketch state.

The size of the sketch is a crucial constraint, and we denote the space consumption in bits by $b$. For example, FM85 consists of $k$ bitmaps of length $\ell$. Thus, its state $s \in \Omega = \{0, 1\}^{k \times \ell}$. Typically, $\ell = 32$, so that $b = 32k$. Further examples are given in Section 4. Our goal is to prove such sketches are differentially private.

## 2 Hash-Based Order-Invariant Estimators are Private

The distribution of any hash-based, order-invariant cardinality estimator depends only on the cardinality of the input stream, so without loss of generality we assume the input is $\mathcal{D} = \{1, \dots, n\}$. Denote the set $\mathcal{D} \backslash \{i\}$ by $\mathcal{D}_{-i}$ for $i \in \mathcal{D}$ and a sketching algorithm with internal randomness $r$ by $S_r(\mathcal{D})$.

By definition, for an $\epsilon$-differential privacy guarantee, we must show that the Bayes factor comparing the hypothesis $i \in \mathcal{D}$ versus $i \notin \mathcal{D}$ is appropriately bounded:

$$e^{-\epsilon} < \frac{\Pr_r(S_r(\mathcal{D}) = s)}{\Pr_r(S_r(\mathcal{D}_{-i}) = s)} < e^{\epsilon} \quad \forall s \in \Omega, i \in \mathcal{D}. \tag{1}$$

**Overview of privacy results.** The main result in our analysis bounds the privacy loss of a hash-based, order-invariant sketch in terms of just two sketch-specific quantities. Both quantities intuitively capture how sensitive the sketch is to the removal or insertion of a single item from the data stream.

The first quantity is a bound $k_{max}$ on the number of items that would change the sketch if *removed* from the stream. Denote the items whose removal from the data set changes the sketch by

$$\mathcal{K}_r := \{i \in \mathcal{D} : S_r(\mathcal{D}_{-i}) \neq S_r(\mathcal{D})\}. \tag{2}$$

Denote its cardinality by $K_r := |\mathcal{K}_r|$ and the upper bound by $k_{max} = \sup_r K_r$.

The second quantity is a bound on a "sampling" probability. Let $\pi(s)$ be the probability that a newly *inserted* item would change a sketch in state $s$,

$$\pi(s) := \Pr_r(S_r(\mathcal{D}) \neq S_r(\mathcal{D}_{-i}) \,|\, S_r(\mathcal{D}_{-i}) = s). \tag{3}$$

Although a sketch generally does not store explicit samples, conceptually, it can be helpful to think of $\pi(s)$ as the probability that an as-yet-unseen item $i$ gets "sampled" by a sketch in state $s$. We upper bound $\pi^* := \sup_{s \in \Omega} \pi(s)$ to limit the influence of items added to the stream.

The main sub-result in our analysis (Theorem 2.4) roughly states that the sketch is $\epsilon$-DP so long as (a) the sampling probability $\pi^* < 1 - e^{-\epsilon}$ is small enough, and (b) the stream cardinality $n > \frac{k_{max}}{e^{\epsilon}-1} = \Theta(k_{max}/\epsilon)$ is large enough.

We show Property (a) is a *necessary* condition for any $\epsilon$-DP algorithm if the algorithm works over data universes of unbounded size. Unfortunately, Property (a) does *not* directly hold for natural

---

[2]A sketch is *duplication-invariant* if and only if its state when run on any stream $\sigma$ is identical to its state when run on the stream $\sigma'$, in which all elements of the stream $\sigma$ appear exactly once.

sketching algorithms. But we show (Section 2.2) by applying a simple down-sampling procedure, any hash-based, order-invariant algorithm can be modified to satisfy (a).

Furthermore, Section 4 shows common sketches satisfy Property (a) with high probability, thus providing $(\epsilon, \delta)$-DP guarantees for sufficiently large cardinalities. Compared to [5], these guarantees are tighter, more precise, and more general as they establish the failure probability $\delta$ decays exponentially with $n$, provide explicit formulas for $\delta$, and apply to a range of sketches rather than just HLL.

**Overview of the analysis.**    The definition of $\epsilon$-DP requires bounding the Bayes factor in equation 1. The challenge is that the numerator and denominator may not be easy to compute by themselves. However, it is similar to the form of a conditional probability involving only one insertion. Our main trick re-expresses this Bayes factor as a sum of conditional probabilities involving a single insertion. Since the denominator $\Pr_r(S_r(\mathcal{D}_{-i}) = s)$ involves a specific item $i$ which may change the sketch, we instead consider the smallest item $J_r$ whose removal does not change the sketch. This allows us to re-express the numerator in terms of a conditional probability $\Pr_r(S(\mathcal{D}) = s \wedge J_r = j) = \Pr_r(J_r = j | S(\mathcal{D}_{-j}) = s) \Pr_r(S(\mathcal{D}_{-j}) = s)$ involving only a single insertion plus a nuisance term $\Pr_r(S(\mathcal{D}_{-j}) = s)$. The symmetry of items gives that the nuisance term is equal to denominator $\Pr_r(S(\mathcal{D}_{-j}) = s) = \Pr_r(S(\mathcal{D}_{-i}) = s)$, thus allowing us to eliminate it.

**Lemma 2.1.** *Suppose $n > \sup_r K_r$. Then $\Pr_r(K_r = n) = 0$, and*

$$\frac{\Pr_r(S_r(\mathcal{D}) = s)}{\Pr_r(S_r(\mathcal{D}_{-i}) = s)} = \sum_{j \in \mathcal{D}} \Pr_r(J_r = j \mid S_r(\mathcal{D}_{-j}) = s). \tag{4}$$

By further conditioning on the total number of items that, when removed, can change the sketch, we obtain conditional probabilities that are simple to calculate. A combinatorial argument simplifies the resulting expression and gives us two factors in Lemma 2.2, one involving the sampling probability for new items $\pi(s)$ given a sketch in state $s$ and the other being an expectation involving $K_r$. This identifies the two quantities that must be controlled in order for a sketch to be $\epsilon$-DP.

**Lemma 2.2.** *Under the same assumptions as Lemma 2.1*

$$\sum_{j \in \mathcal{D}} \Pr_r(J_r = j \mid S_r(\mathcal{D}_{-j}) = s) = (1 - \pi(s)) \mathbb{E}_r \left( 1 + \frac{K_r}{n - K_r + 1} \bigg| S_r(\mathcal{D}_{-1}) = s \right). \tag{5}$$

To show that all hash-based, order invariant sketching algorithms can be made $\epsilon$-DP, we show that $K_r$ can always be bounded by the maximum size of the sketch in bits. Thus, if a sketch is combined with a downsampling procedure to ensure $\pi(s)$ is sufficiently small, one satisfies both of the properties that are sufficient for an $\epsilon$-DP guarantee.

Having established (5), we can derive a result showing that a hash-based, order-invariant sketch is $\epsilon$-DP so long as the stream cardinality is large enough and $\sup_{s \in \Omega} \pi(s)$ is not too close to 1.

**Corollary 2.3.** *Let $\Omega$ denote the set of all possible states of a hash-based order-invariant distinct counting sketching algorithm. When run on a stream of cardinality $n > \sup_r K_r$, the sketch output by the algorithm satisfies $\epsilon$-DP if*

$$\pi_0 := 1 - e^{-\epsilon} > \sup_{s \in \Omega} \pi(s) \quad and \tag{6}$$

$$e^\epsilon > 1 + \mathbb{E}_r \left( \frac{K_r}{n - K_r + 1} \bigg| S_r(\mathcal{D}_{-1}) = s \right) \quad for\ all\ sketch\ states\ s \in \Omega. \tag{7}$$

*Furthermore, if the data stream $\mathcal{D}$ consists of items from a universe $U$ of unbounded size, Condition 6 is necessarily satisfied by* any *sketching algorithm satisfying $\epsilon$-DP.*

The above corollary may be difficult to apply directly since the expectation in Condition (7) is often difficult to compute and depends on the unknown cardinality $n$. Our main result provides sufficient criteria to ensure that Condition (7) holds. The criteria is expressed in terms of a minimum cardinality $n_0$ and sketch-dependent constant $k_{max}$. This constant $k_{max}$ is a bound on the maximum number of items which change the sketch when removed. That is, for all input streams $\mathcal{D}$ and all $r$, $k_{max} \geq |\mathcal{K}_r|$. We derive $k_{max}$ for a number of popular sketch algorithms in Section 4.

**Theorem 2.4.** *Consider any hash-based, order-invariant distinct counting sketch. The sketch output by the algorithm satisfies an $\epsilon$-DP guarantee if*

$$\sup_{s\in\Omega} \pi(s) < \pi_0 := 1 - e^{-\epsilon} \quad \text{and there are strictly greater than} \tag{8}$$

$$n_0 := k_{max}/(1 - e^{-\epsilon}) \quad \text{unique items in the stream.} \tag{9}$$

Later, we explain how to modify existing sketching algorithms in a black-box way to satisfy these conditions. If left unmodified, most sketching algorithms used in practice allow for some sketch values $s \in \Omega$ which violate Condition 8, i.e $\pi(s) > 1 - e^{-\epsilon}$. We call such sketch values "privacy-violating". Fortunately, such values turn out to arise with only tiny probability. The next theorem states that, so long as this probability is smaller than $\delta$, the sketch satisfies $(\epsilon, \delta)$-DP without modification. The proof of Theorem 2.5 follows immediately from Theorem 2.4.

**Theorem 2.5.** *Let $n_0$ be as in Theorem 2.4. Given a hash-based, order-invariant distinct counting sketch with bounded size, let $\Omega'$ be the set of sketch states such that $\pi(s) \geq \pi_0$. If the input stream $\mathcal{D}$ has cardinality $n > n_0$, then the sketch is $(\epsilon, \delta)$ differentially private where $\delta = \Pr_r(S_r(\mathcal{D}) \in \Omega')$.*

## 2.1 Constructing Sketches Satisfying Approximate Differential Privacy: Algorithm 1a

Theorem 2.5 states that, when run on a stream with $n \geq n_0$ distinct items, any hash-based order-invariant algorithm (see Algorithm 1a) automatically satisfies $(\epsilon, \delta)$-differential privacy where $\delta$ denotes the probability that the final sketch state $s$ is "privacy-violating", i.e., $\pi(s) > \pi_0 = 1 - e^{-\epsilon}$. In Section 4, we provide concrete bounds of $\delta$ for specific algorithms. In all cases considered, $\delta$ falls exponentially with respect to the cardinality $n$. Thus, high privacy is achieved with high probability so long as the stream is large.

We now outline how to derive a bound for a specific sketch. We can prove the desired bound on $\delta$ by analyzing sketches in a manner similar to the coupon collector problem. Assuming a perfect, random hash function, the hash values of a universe of items defines a probability space. We can identify $v \leq k_{max}$ events or coupons, $C_1, \ldots, C_v$, such that $\pi(s)$ is guaranteed to be less than $\pi_0$ after all events have occurred. Thus, if all coupons are collected, the sketch satisfies the requirement to be $\epsilon$-DP. As the cardinality $n$ grows, the probability that a particular coupons remains missing decreases exponentially. A simple union bound shows that the probability $\delta$ that *any* coupon is missing decreases exponentially with $n$.

For more intuition as to why unmodified sketches satisfy an $(\epsilon, \delta)$-DP guarantee when the cardinality is large, we note that the inclusion probability $\pi(s)$ is closely tied to the cardinality estimate in most sketching algorithms. For example, the cardinality estimators used in HLL and KMV are inversely proportional to the sampling probability $\pi(s)$, i.e., $\hat{N}(s) \propto 1/\pi(s)$, while for LPCA and Adaptive Sampling, the cardinality estimators are monotonically decreasing with respect to $\pi(s)$. Thus, for most sketching algorithms, when run on a stream of sufficiently large cardinality, the resulting sketch is privacy-violating only when the cardinality estimate is also inaccurate. Theorem 2.6 is useful when analyzing the privacy of such algorithms, as it characterizes the probability $\delta$ of a "privacy violation" in terms of the probability the returned estimate, $\hat{N}(S_r(\mathcal{D}))$, is lower than some threshold $\tilde{N}(\pi_0)$.

**Theorem 2.6.** *Let $S_r$ be a sketching algorithm with estimator $\hat{N}(S_r)$. If $n \geq n_0$ and the estimate returned on sketch $s$ is a strictly decreasing function of $\pi(s)$, so that $\hat{N}(s) = \tilde{N}(\pi(s))$ for a function $\tilde{N}$. Then, $S_r$ is $(\epsilon, \delta)$-DP where $\delta = \Pr_r(\hat{N}(S_r(\mathcal{D})) < \tilde{N}(\pi_0))$.*

## 2.2 Constructing Sketches Satisfying Pure Differential Privacy: Algorithm 1b - 1c

Theorem 2.4 guarantees an $\epsilon$-DP sketch if (8), (9) hold. Condition (8) requires that $\sup_{s\in\Omega} \pi(s) < 1 - e^{-\epsilon}$, i.e., the "sampling probability" of the sketching algorithm is sufficiently small regardless of the sketch's state $s$. Meanwhile, (9) requires that the input cardinality is sufficiently large.

We show that *any* hash-based, order-invariant distinct counting sketching algorithm can satisfy these two conditions by adding a simple pre-processing step which does two things. First, it "downsamples" the input stream by hashing each input, interpreting the hash values as numbers in $[0, 1]$, and simply ignoring numbers whose hashes are larger than $\pi_0$. The downsampling hash must be independent to that used by the sketching algorithm itself. This ensures that Condition (8) is satisfied, as each input item has maximum sampling probability $\pi_0$.

| BASE(items, $\epsilon$) | DPSKETCHLARGESET(items, $\epsilon$) | DPSKETCHANYSET(items, $\epsilon$) |
|---|---|---|
| $S \leftarrow InitSketch()$ | $S \leftarrow InitSketch()$ | $S, n_0 \leftarrow DPInitSketch(\epsilon)$ |
| | $\pi_0 \leftarrow 1 - e^{-\epsilon}$ | $\pi_0 \leftarrow 1 - e^{-\epsilon}$ |
| **for** $x \in items$ **do** | **for** $x \in items$ **do** | **for** $x \in items$ **do** |
| | **if** $hash(x) < \pi_0$ **then** | **if** $hash(x) < \pi_0$ **then** |
| $S.add(x)$ | $S.add(x)$ | $S.add(x)$ |
| **return** $\hat{N}(S)$ | **return** $\hat{N}(S)/\pi_0$ | **return** $\hat{N}(S)/\pi_0 - n_0$ |
| (a) $(\epsilon, \delta)$-DP for $n \geq n_0$. | (b) $(\epsilon, 0)$-DP for $n \geq n_0$. | (c) $(\epsilon, 0)$-DP for $n \geq 1$. |

Algorithms 1: Differentially private cardinality estimation algorithms from black box sketches. The function $InitSketch()$ initializes a black-box sketch. The uniform random hash function $hash(x)$ is chosen independently of any hash in the black-box sketch and is interpreted as a real in $[0, 1]$. The cardinality estimate returned by sketch $S$ is denoted $\hat{N}(S)$. DPInitSketch is given in Algorithm 2a.

If there is an a priori guarantee that the number of distinct items $n$ is greater than $n_0 = \frac{k_{max}}{1-e^{-\epsilon}}$, then (9) is trivially satisfied. Pseudocode for the resulting $\epsilon$-DP algorithm is given in Algorithm 1b. If there is no such guarantee, then the preprocessing step adds $n_0$ items to the input stream to satisfy (9). To ensure unbiasedness, these $n_0$ items must (i) be distinct from any items in the "real" stream, and (ii) be downsampled as per the first modification. An unbiased estimate of the cardinality of the unmodified stream can then be easily recovered from the sketch via a post-processing correction. Pseudocode for the modified algorithm, which is guaranteed to satisfy $\epsilon$-DP, is given in Algorithm 1c.

**Corollary 2.7.** *The functions* `DPSketchLargeSet` *(Algorithm 1b) and* `DPSketchAnySet` *(Algorithm 1c) yield $\epsilon$-DP distinct counting sketches provided that $n \geq n_0$ and $n \geq 1$, respectively.*

### 2.3   Constructing $\epsilon$-DP Sketches from Existing Sketches: Algorithm 3, Appendix A.1

As regulations change and new ones are added, existing data may need to be appropriately anonymized. However, if the data has already been sketched, the underlying data may no longer be available, and even if it is retained, it may be too costly to reprocess it all. Our theory allows these sketches to be directly converted into differentially private sketches when the sketch has a merge procedure. Using the merge procedure to achieve $\epsilon$-differential privacy yields more useful estimates than the naive approach of simply adding Laplace noise to cardinality estimates in proportion to the global sensitivity.

The algorithm assumes it is possible to take a sketch $S_r(\mathcal{D}_1)$ of a stream $\mathcal{D}_1$ and a sketch $S_r(\mathcal{D}_2)$ of a stream $\mathcal{D}_2$, and "merge" them to get a sketch of the concatenation of the two streams $\mathcal{D}_1 \circ \mathcal{D}_2$. This is the case for most practical hash-based order-invariant distinct count sketches. Denote the merge of sketches $S_r(\mathcal{D}_1)$ and $S_r(\mathcal{D}_2)$ by $S_r(\mathcal{D}_1) \cup S_r(\mathcal{D}_2)$. In this setting, we think of the existing non-private sketch $S_r(\mathcal{D}_1)$ being converted to a sketch that satisfies $\epsilon$-DP by Algorithm 3 (see pseudocode in Appendix A.1). Since sketch $S_r(\mathcal{D}_1)$ is already constructed, items cannot be first downsampled in the build phase the way they are in Algorithms 1b-1c. To achieve $\epsilon$-DP, Algorithm 3 constructs a noisily initialized sketch, $S_r(\mathcal{D}_2)$, which satisfies both the downsampling condition (Condition (8)) and the minimum stream cardinality requirement (Condition (9)) and returns the merged sketch $S_r(\mathcal{D}_1) \cup S_r(\mathcal{D}_2)$. Hence, the sketch will satisfy both conditions for $\epsilon$-DP, as shown in Corollary A.3

This merge based procedure typically adds no additional error to the estimates for large cardinalities. In contrast, the naive approach of adding Laplace noise can add significant noise since the sensitivity can be very large. For example, HLL's estimator is of the form $\hat{N}_{HLL}(s) = \alpha/\pi(s)$ where $\alpha$ is a constant and $s$ is the sketch. One item can update a bin to the maximum value, so that the updated sketch $s'$ has sampling probability $\pi(s') < \pi(s)(1 - 1/k)$. The sensitivity of cardinality estimate is thus at least $\hat{N}_{HLL}(s)/k$. Given that the cardinality estimate, and hence sensitivity, can be arbitrarily large when $n \geq k$, the naive approach is unworkable to achieve $\epsilon$-DP.

## 3   The Utility of Private Sketches

When processing a data set with $n$ unique items, denote the expectation and variance of a sketch and its estimator by $\mathbb{E}_n(\hat{N})$ and $\text{Var}_n(\hat{N})$ respectively. We show that our algorithms all yield unbiased

estimates. Furthermore, we show that for Algorithms 1a-1c, if the base sketch satisfies a *relative error guarantee* (defined below), the DP sketches add no additional error asymptotically.

**Establishing unbiasedness.** To analyze the expectation and variance of each algorithm's estimator, $\hat{N}(S(\mathcal{D}))$, note that each estimator uses a 'base estimate' $\hat{N}_{base}$ from the base sketch $S$ and has the form $\hat{N}(S(\mathcal{D})) = \frac{\hat{N}_{base}}{p} - V$; $p$ is the downsampling probability and $V$ is the number of artificial items added. This allows us to express expectations and variance via the variance of the base estimator.

**Theorem 3.1.** *Consider a base sketching algorithm $S \in \mathcal{C}$ with an unbiased estimator $\hat{N}_{base}$ for the cardinality of items added to the base sketch. Algorithms 1 (a)-(c) and 3 yield unbiased estimators.*

**Bounding the variance.** Theorem 3.1 yields a clean expression for the variance of our private algorithms. Namely, $\mathsf{Var}[\hat{N}(S_r(\mathcal{D}))] = \mathbb{E}[\mathsf{Var}(\frac{\hat{N}_{base}}{p}|V)]$ which is shown in Corollary B.1. The expression is a consequence of the law of total variance and that the estimators are unbiased.

We say that the base sketch satisfies a **relative-error guarantee** if with high probability, the estimate returned by the sketching algorithm when run on a stream of cardinality $n$ is $(1 \pm 1/\sqrt{c})n$ for some constant $c > 0$. Let $\hat{N}_{base,n}$ denote the cardinality estimate when the base algorithm is run on a stream of cardinality $n$, as opposed to $\hat{N}_{base}$ denoting the cardinality estimate produced by the base sketch on the sub-sampled stream used in our private sketches DPSketchLargeSet (Algorithm 1b) and DPSketchAnySet (Algorithm 1c). The relative error guarantee is satisfied when $\mathsf{Var}_n(\hat{N}_{base,n}) < n^2/c$; this is an immediate consequence of Chebyshev's inequality.

When the number of artificially added items $V$ is constant as in Algorithms 1b and 1c, Corollary B.1 provides a precise expression for the variance of the differentially private sketch. In Theorem 3.2 below, we use this expression to establish that the modification of the base algorithm to an $\epsilon$-DP sketch as per Algorithms 1b and 1c satisfy the exact same relative error guarantee asymptotically. In other words, the additional error due to any pre-processing (down-sampling and possibly adding artificial items) is insignificant for large cardinalities $n$.

**Theorem 3.2.** *Suppose $\hat{N}_{base,n}$ satisfies a relative error guarantee, $\mathsf{Var}_n(\hat{N}_{base,n}) < n^2/c$, for all $n$ and for some constant c. Let $v = 0$ for Algorithm 1b and $v = n_0$ for Algorithm 1c. Then Algorithms 1b and 1c satisfy*

$$\mathsf{Var}_n(\hat{N}) \leq \frac{(n+v)^2}{c} + \frac{(n+v)(v+\pi_0^{-1})}{k_{max}} = \frac{(n+v)^2}{c} + O(n), \tag{10}$$

*so that $\mathsf{Var}_n(\hat{N})/\mathsf{Var}_n(\hat{N}_{base,n}) \to 1$ as $n \to \infty$.*

In Corollary B.2 we prove an analogous result for Algorithm 3, which merges non-private and noisy sketches to produce a private sketch. Informally, the result is comparable to (10), albeit with $v \geq n_0$. This is because, in Algorithm 3, the number of artificial items added $V$ is a random variable. We ensure that the algorithm satisfies a utility guarantee by bounding $V$ with high probability. This is equivalent to showing that the base sketching algorithm satisfies an $(\epsilon, \delta)$-DP guarantee as for any $n^* \geq n_0$ and dataset $\mathcal{D}^*$ with $|\mathcal{D}^*| = n^*$, $(\epsilon, \delta_{n^*})$-DP ensures $\delta_{n^*} > \Pr_r(\pi(S_r(\mathcal{D}^*)) > \pi_0) = \Pr_r(V > n^*)$ which follows from the definition of $V$ in Algorithm 2b.

## 4 Examples of Hash-based, Order-Invariant Cardinality Estimators

We now provide $(\epsilon, \delta)$-DP results for a select group of samples: FM85, LPCA, Bottom-$k$, Adaptive Sampling, and HLL. The $(\epsilon, \delta)$-DP results in this section operate in the Algorithm 1a setting with no modification to the base sketching algorithm. Recall that the quantities of interest are the number of bins used in the sketch $k$, the size of the sketch in bits $b$ and the number of items whose absence changes the sketch $k_{max}$. From Section 2 and Lemma A.1 we know that $k_{max} \leq b$ but for several common sketches we show a stronger bound of $k_{max} = k$. The relationship between these parameters for various sketching algorithms is summarized in Table 1. Table 2, Appendix C, details our improvements over [22, 5] in both privacy and utility.

We remind the reader that, per (6), $\pi_0 = 1 - e^{-\epsilon}$, and (9) $n_0 = \frac{k_{max}}{1-e^{-\epsilon}}$. Furthermore, recall that once we bound the parameter $k_{max}$ for any given hash-based order-invariant sketching algorithm,

Table 1: Properties of each sketch with $k$ "buckets" (see each sketch's respective section for details of what this parameter means for the sketch). Each sketch provides an $(\epsilon, \delta)$-DP guarantee, where the column $\ln \delta$ provides an upper bound on $\ln \delta$ established in the relevant subsection of Section 4.

| Sketch | $b$: size (bits) | Standard Error | $k_{max}$ | $\ln \delta$ | Reference |
|---|---|---|---|---|---|
| FM85 | $32k$ | $0.649n/\sqrt{k}$ | $32k$ | $-\frac{\pi_0}{2k}n + o(1)$ | [16] |
| LPCA | $k$ | $n/\sqrt{k}$ [3] | $k$ | $-\frac{\pi_0}{\tilde{N}(\pi_0)}n + O(\log n)$ | [27] |
| Bottom-$k$ | $64k$ | $n/\sqrt{k}$ | $k$ | $-\frac{1}{2}\frac{\pi_0}{1-\pi_0}n + o(1)$ | [13] |
| Adaptive Sampling | $k$ | $1.2n/\sqrt{k}$ | $k$ | $-\frac{1}{2}\frac{\pi_0}{1-\pi_0}n + o(1)$ | [10] |
| HLL | $5k$ | $1.04\,n/\sqrt{k}$ | $k$ | $-\frac{\pi_0}{k}n + o(1)$ | [11] |

Corollary 2.7 states that the derived algorithms 1b-1c satisfy $\epsilon$-DP provided that $n \geq n_0$ and $n \geq 1$, respectively. Accordingly, in the rest of this section, we bound $k_{max}$ for each example sketch of interest, which has the consequences for pure $\epsilon$-differential privacy delineated above.

**Flajolet-Martin '85** The FM85 sketch, often called *Probabilistic Counting with Stochastic Averaging (PCSA)*, consists of $k$ bitmaps $B_i$ of length $\ell$. Each item is hashed into a bitmap and index $(B_i, G_i)$ and sets the indexed bit in the bitmap to 1. The chosen bitmap is uniform amongst the $k$ bitmaps and the index $G_i \sim Geometric(1/2)$. If $\ell$ is the length of each bitmap, then the total number of bits used by the sketch is $b = k\ell$ and $k_{max} = k\ell$ for all seeds $r$. A typical value for $\ell$ is 32 bits, as used in Table 1. Past work [25] proposed an $\epsilon$-DP version of FM85 using a similar subsampling idea combined with random bit flips.

**Theorem 4.1.** *Let $v = \lceil -\log_2 \pi_0 \rceil$ and $\tilde{\pi}_0 := 2^{-v} \in (\pi_0/2, \pi_0]$. If $n \geq n_0$, then the FM85 sketch is $(\epsilon, \delta)$-DP with $\delta \leq kv \exp\left(-\tilde{\pi}_0 \frac{n}{k}\right)$.*

For any $k$, FM85 has $k_{max} \in \{32k, 64k\}$. This is worse than all other sketches we study which have $k_{max} = k$, so FM85 needs a larger number of minimum items $n_0$ to ensure the sketch is $(\epsilon, \delta)$-DP.

**LPCA** The Linear Probabilistic Counting Algorithm (LPCA) consists of a length-$k$ bitmap. Each item is hashed to an index and sets its bit to 1. If $B$ is the number of 1 bits, the LPCA cardinality estimate is $\hat{N}_{\text{LPCA}} = -k \log(1 - B/k) = k \log \pi(S_r(\mathcal{D}))$. Trivially, $k_{max} = k$.

Since all bits are expected to be 1 after processing roughly $k \log k$ distinct items, the capacity of the sketch is bounded. To estimate larger cardinalities, one first downsamples the distinct items with some sampling probability $p$. To ensure the sketch satisfies an $\epsilon$-DP guarantee, one simply ensures $p \geq \pi_0$. In this case, our analysis shows that LPCA is differentially private with no modifications if the cardinality is sufficiently large. Otherwise, since the estimator $\hat{N}(s)$ is a function of the sampling probability $\pi(s)$, Theorem 2.6 provides an $(\epsilon, \delta)$ guarantee in terms of $\hat{N}$.

**Theorem 4.2.** *Consider a LPCA sketch with $k$ bits and downsampling probability $p$. If $p < \pi_0$ and $n > \frac{k}{1-e^{-\epsilon}}$ then LPCA is $\epsilon$-DP. Otherwise, let $b_0 = \lceil k(1 - \pi_0/p) \rceil$, $\tilde{\pi}_0 = b_0/k$, and $\mu_0$ be the expected number of items inserted to fill $b_0$ bits in the sketch. Then, LPCA is $(\epsilon, \delta)$-DP if $n > \mu_0$ with*

$$\delta = \Pr_r(B < b_0) < \frac{\mu_0}{n} \exp\left(-\frac{\tilde{\pi}_0}{\mu_0}n\right) \exp(-\tilde{\pi}_0) \tag{11}$$

*where $B$ is the number of filled bits in the sketch. Furthermore, $\mu_0 < \tilde{N}(\tilde{\pi}_0)$ where $\tilde{N}(\tilde{\pi}) = -\frac{k}{p} \log(1 - \tilde{\pi})$ is the cardinality estimate of the sketch when the sampling probability is $\tilde{\pi}$.*

**Bottom-$k$ (also known as MinCount or KMV)** sketches store the $k$ smallest hash values. Removing an item changes the sketch if and only if 1) the item's hash value is one of these $k$ and 2) it does not collide with another item's hash value. Thus, $k_{max} = k$. Typically, the output size of the hash function is large enough to ensure that the collision probability is negligible, so for practical purposes $k_{max} = k$ exactly. Since the Bottom-$k$ estimator $\hat{N}(s) = {(k-1)}/{\pi(s)}$ is a function of the update probability $\pi(s)$, Theorem 2.6 gives an $(\epsilon, \delta)$-DP guarantee in terms of the cardinality estimate by coupon collecting; Theorem 4.3 tightens this bound on $\delta$ for a stronger $(\epsilon, \delta)$-DP guarantee.

---

[3]This approximation holds for $n < k$. A better approximation of the error is $\sqrt{k(\exp(n/k) - n/k - 1)}$

**Theorem 4.3.** *Consider Bottom-$k$ with $k$ minimum values. Given $\epsilon > 0$, let $\pi_0, n_0$ be the corresponding subsampling and minimum cardinality to ensure the modified Bottom-$k$ sketch is $(\epsilon, 0)$-DP. When run on streams of cardinality $n \geq n_0$, then the unmodified sketch is $(\epsilon, \delta)$-DP, where $\delta = P(X \leq k) < \exp(-n\alpha_n)$ where $X \sim Binomial(n, \pi_0)$ and $\alpha_n = \frac{1}{2} \frac{(\pi_0 - k/n)^2}{\pi_0(1-\pi_0) + 1/3n^2} \to \frac{1}{2} \frac{\pi_0}{1-\pi_0}$ as $n \to \infty$.*

The closely related **Adaptive Sampling** sketch has the same privacy behavior as a bottom-$k$ sketch. Rather than storing exactly $k$ hashes, the algorithm maintains a threshold $p$ and stores up to $k$ hash values beneath $p$. Once the sketch size exceeds $k$, the threshold is halved and only hashes less than $p/2$ are kept. Since at most $k$ hashes are stored, and the sketch is modified only if one of these hashes is removed the maximum number of items that can modify the sketch by removal is $k_{max} = k$.

**Corollary 4.4.** *For any size $k$ and cardinality $n$, if a bottom-$k$ sketch is $(\epsilon, \delta)$-DP, then a maximum size $k$ adaptive sampling sketch is $(\epsilon, \delta)$-DP with the same $\epsilon$ and $\delta$.*

**HyperLogLog (HLL)**   hashes each item to a bin and value $(B_i, G_i)$. Within each bin, it takes the maximum value so each bin is a form of Bottom-1 sketch. If there are $k$ bins, then $k_{max} = k$.

Our results uniformly improve upon existing DP results on the HLL sketch and its variants. One variation of the HLL sketch achieves $\epsilon$-DP but is far slower than HLL, as it requires every item to be independently hashed once for each of the $k$ bins, rather than just one time [22]. In other words, [22] needs $O(k)$ update time compared to $O(1)$ for our algorithms. Another provides an $(\epsilon, \delta)$ guarantee for streams of cardinality $n \geq n_0'$, for an $n_0'$ that is larger than our $n_0$ by a factor of roughly (at least) 8, with $\delta$ falling exponentially with $n$ [5]. In contrast, for streams with cardinality $n \geq n_0$, we provide a *pure* $\epsilon$-DP guarantee using Algorithms 1b-1c. HLL also has the following $(\epsilon, \delta)$ guarantee.

**Theorem 4.5.** *If $n \geq n_0$, then HLL satisfies an $(\epsilon, \delta)$-DP guarantee where $\delta \leq k \exp(-\pi_0 n/k)$*

HLL's estimator is only a function of $\pi(s)$ for medium to large cardinalities as it has the form $\hat{N}(s) = \tilde{N}(\pi(s))$ when $\tilde{N}(\pi(s)) > 5k/2$. Thus, if $\pi_0$ is sufficiently small so that $\tilde{N}(\pi_0(s)) > 5k/2$, then Theorem 2.6 can still be applied, and HLL satisfies $(\epsilon, \delta)$-DP with $\delta = P(\hat{N}(S_r(\mathcal{D})) < \tilde{N}(\pi_0))$.

## 5   Empirical Evaluation

We provide two experiments highlighting the practical benefits of our approach. Of past works, only [5, 22] are comparable and both differ from our approach in significant ways. We empirically compare only to [22] since [5] is simply an analysis of HLL. Our improvement over [5] for HLL consists of providing significantly tighter privacy bounds in Section 4 and providing a fully $\epsilon$-DP sketch in the secret hash setting. We denote our $\epsilon$-DP version of HLL using Algorithm 1b by PHLL (private-HLL) and that of [22] by QLL. Details of the experimental setup are in Appendix D.

**Experiment 1: Update Time (Figure 1a).** We implemented regular, non-private HLL, our PHLL, and QLL and recorded the time to populate every sketch over $2^{10}$ updates with $k \in \{2^7, 2^8, \dots 2^{12}\}$ buckets. For HLL, these bucket sizes correspond to relative standard errors ranging from $\approx 9\%$ down to $\approx 1.6\%$. Each marker represents the mean update time over all updates and the curves are the evaluated mean update time over 10 trials.

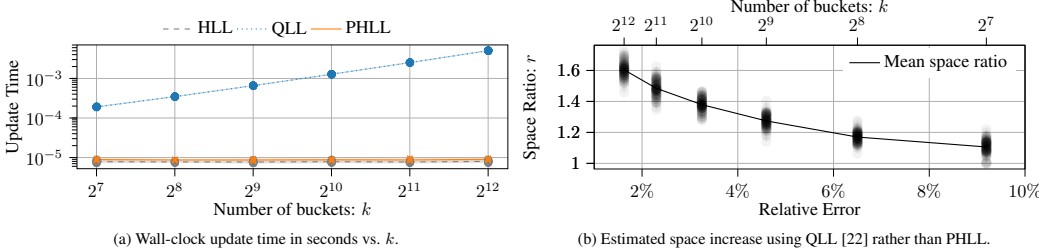

(a) Wall-clock update time in seconds vs. $k$.

(b) Estimated space increase using QLL [22] rather than PHLL.

Figure 1: (1a) QLL's update time is not competitive since it performs $O(k)$ hashes. (1b) QLL is less efficient spacewise than PHLL. The relative size of a QLL sketch to a PHLL sketch, the *Space ratio*, is larger for more accurate sketches.

As expected from theory, the update time of [22] grows as $O(k)$. In contrast, our method PHLL has a constant update time and is similar in magnitude to HLL. Both are roughly $500\times$ faster than [22] when $k = 2^{12}$. Thus, figure 1a shows that [22] is not a scalable solution and the speedup by achieving $O(1)$ updates is substantial.

**Experiment 2: Space Comparison (Figure 1b).** In addition to having a worse update time, we also show that QLL has lower utility in the sense that it requires more space than PHLL to achieve the same error. Fixing the input cardinality at $n = 2^{20}$ and the privacy budget at $\epsilon = \ln(2)$, we vary the number of buckets $k \in \{2^7, 2^8, \dots 2^{12}\}$ and simulate the $\epsilon$-DP methods, PHLL and QLL [22]. The number of buckets controls the error and we found that both methods obtained very similar mean relative error for a given number of bins[4] so we plot the space usage against the expected relative error for a given number of buckets. For QLL, since the error guarantees tie the parameter $\gamma$ to the number of buckets, we modify $\gamma$ accordingly as well. We compare the sizes of each sketch as the error varies.

Since the number of bits required for each bin depends on the range of values the bin can take, we record the simulated **total sketch size** $:= k \cdot \log_2 \max_i s_i$, by using the space required for the largest bin value over $k$ buckets.

Although QLL achieves similar utility, it does so using a sketch that is larger: when $k = 2^7$, we expect an error of roughly $9\%$, QLL is roughly $1.1\times$ larger. This increases to about $1.6\times$ larger than our PHLL sketch when $k = 2^{12}$, achieving error of roughly $1.6\%$. We see that the average increase in space when using QLL compared to PHLL *grows exponentially in the desired accuracy of the sketch*; when lower relative error is necessary, we obtain a greater space improvement over QLL than at higher relative errors. This supports the behavior expected by comparing with space bounds of [22] with (P)HLL.

# 6 Conclusion

We have studied the (differential) privacy of a class of cardinality estimation sketches that includes most popular algorithms. Two examples are the HLL and KMV (bottom-$k$) sketches that have been deployed in large systems [14, 1]. We have shown that the sketches returned by these algorithms are $\epsilon$-differentially private when run on streams of cardinality greater than $n_0 = \frac{k_{max}}{1 - e^{-\epsilon}}$ and when combined with a simple downsampling procedure. Moreover, even without downsampling, these algorithms satisfy $(\epsilon, \delta)$-differential privacy where $\delta$ falls exponentially with the stream cardinality $n$ once $n$ is larger than the threshold $n_0$. Our results are more general and yield better privacy guarantees than prior work for small space cardinality estimators that preserve differential privacy. Our empirical validations show that our approach is practical and scalable, being much faster than previous state-of-the-art while consuming much less space.

## Acknowledgments and Disclosure of Funding

We are grateful to Graham Cormode for valuable comments on an earlier version of this manuscript. Justin Thaler was supported by NSF SPX award CCF-1918989 and NSF CAREER award CCF-1845125.

---

[4]This is shown in Figure 3, Appendix D.

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
