DPINITSKETCH($\epsilon$)
 $S \leftarrow InitSketch()$
 $\pi_0 \leftarrow 1 - e^{-\epsilon}$
 $n_0 \leftarrow \left\lceil \frac{k_{max}-1}{\pi_0} \right\rceil$
 $M \sim Binomial(n_0, \pi_0)$
 **for** $i = 1 \rightarrow M$ **do**
  $x \leftarrow NewItem()$
  **if** $hash(x) < \pi_0$ **then**
   $S.add(x)$

 **return** $S, n_0$

(a)

DPINITSKETCHFORMERGE($\epsilon$)
 $S \leftarrow InitSketch()$
 $\pi_0 \leftarrow 1 - e^{-\epsilon}$
 $n_0 \leftarrow \left\lceil \frac{k_{max}-1}{\pi_0} \right\rceil$
 $v \leftarrow 0$
 **repeat**
  $x \leftarrow NewItem()$
  $S.add(x)$
  $v \leftarrow v + 1$
 **until** $\pi(S) \leq \pi_0$ and $v \geq n_0$
 **return** $S, v$

(b)



Algorithms 2: Initialization routines for generating $\epsilon$-DP sketches. The function $NewItem()$ returns an item that is guaranteed to come from a data universe disjoint from the universe over which stream items are drawn. In DPInitSketch, the binomial draw $M$ simulates inserting $n_0$ unique items into the sketch, with downsampling probability $\pi_0$.

## A  Omissions from Section 2

### A.1  Algorithms

Due to space considerations, the algorithms for initializing $\epsilon$-DP sketches has been omitted from the main body. So too has the algorithm for constructing an existing non-private sketch from a private sketch. These are detailed in Algorithms 2 and Algorithm 3, respectively.

---

**Algorithm 3** Turn an existing sketch into one with an $\epsilon$-DP guarantee.

---
 MAKEDP($S, \epsilon$)
  $T, v \leftarrow DPInitSketchForMerge(\epsilon)$          ▷ Algorithm 2b
  **return** $S \cup T, \hat{N}(S \cup T) - v$
      ▷ return private sketch and associated cardinality estimate for stream $S$ is a sketch of.

---

### A.2  Technical Results

Our first result shows that all sketches under consideration cannot be modified by too many items. Specifically, we show that the number of items that can change the sketch, $K_r$, is bounded above by $b$ which is the number of bits required to store the sketch. However, as shown in Section 4, this can in fact be strengthened for many specific sketches.

**Lemma A.1.** *Suppose that* $n > \sup_r K_r$. *Then for any distinct counting sketch with size in bits bounded by* $b$, $\sup_r K_r \leq b$.

**Proof of Lemma A.1.** Consider some data set $\mathcal{D}$ and sketch $s = S_r(\mathcal{D})$. Recall that we denote the set of items whose removal would change the sketch by $\mathcal{K}_r(\mathcal{D}) := \{i \in \mathcal{D} : S_r(\mathcal{D}_{-i}) \neq S_r(\mathcal{D})\}$. Consider any subset $\Lambda \subset \mathcal{K}_r(\mathcal{D})$. Then we claim that, for any $x \in \mathcal{K}_r$, adding $x$ to the sketch $S_r(\Lambda)$ will change it if and only if $x \in \mathcal{K}_r(\mathcal{D}) \setminus \Lambda$. That is, if $\Lambda \circ x$ denotes the stream consisting of one occurrence of each item in $\Lambda$, followed by $x$, then $S_r(\Lambda) \neq S_r(\Lambda \circ x)$ if and only if $x \in \mathcal{K}_r(\mathcal{D}) \setminus \Lambda$.

To see this, first observe that duplication-invariance of the sketching algorithm implies that if $x \in \Lambda$ then $S_r(\Lambda) = S_r(\Lambda \circ x)$. Second if $x \notin \Lambda$, suppose by way of contradiction that $S_r(\Lambda) = S_r(\Lambda \circ x)$, and let $T = \mathcal{D} \setminus (\Lambda \cup \{x\})$. Since $S_r(\Lambda) = S_r(\Lambda \circ x)$, it holds that $S_r(\Lambda \circ x \circ T) = S_r(\Lambda \circ T) = S_r(\mathcal{D}_{-x})$. Yet by order-invariance of the sketching algorithm, $S_r(\Lambda \circ x \circ T) = S_r(\mathcal{D})$. We conclude that $S_r(\mathcal{D}_{-x}) = S_r(\mathcal{D})$, contradicting the assumption that $x \in \mathcal{K}_r(\mathcal{D})$.

The above means that for any fixed $r$, the sketch $S_r(\Lambda)$ losslessly encodes the arbitrary subset $\Lambda$ of $\mathcal{K}_r$. Hence, the sketch requires at least $\log_2(2^{|\mathcal{K}_r|}) = |\mathcal{K}_r|$ bits to represent. Thus, any sketch with size bounded by $m$ bits can have at most $m$ items that affect the sketch. $\square$

Next we prove Lemma 2.1 which expresses the Bayes factor from (1)

$$\frac{\Pr_r(S_r(\mathcal{D}) = s)}{\Pr_r(S_r(\mathcal{D}_{-i}) = s)}$$

to a sum of conditional probabilities involving a single insertion.

**Proof of Lemma 2.1.** Recall from Equation (2) that

$$\mathcal{K}_r := \{i \in \mathcal{D} : S_r(\mathcal{D}_{-i}) \neq S_r(\mathcal{D})\}$$

denotes the set of items that would change the state of the sketch if removed, and its cardinality is $K_r := |\mathcal{K}_r|$. We also define $J_r := \min\{i : S_r(\mathcal{D}_{-i}) = S_r(\mathcal{D})\}$ to be the smallest index amongst the remaining $n - K_r$ items in $\mathcal{D}$ that do not change the sketch. If removing *any* item changes the sketch, i.e., if $S_r(\mathcal{D}_{-i}) \neq S_r(\mathcal{D})$ for all $i \in \mathcal{D}$, then $K_r = n$. For this case, we define $J_r$ to be a special symbol $\perp$.

First, let us rewrite $\Pr_r(S_r(\mathcal{D}) = s)$ as a sum over all possible values of $J_r$:

$$\Pr_r(S_r(\mathcal{D}) = s) = \sum_{j \in \mathcal{D} \cup \{\perp\}} \Pr_r(J_r = j \wedge S_r(\mathcal{D}) = s). \tag{12}$$

Next, we split the right hand side of Equation (12) into the distinct cases wherein $J_r = \perp$ and $j \in \mathcal{D}$, as we will ultimately deal with each case separately:

$$\sum_{j \in \mathcal{D} \cup \{\perp\}} \Pr_r(J_r = j \wedge S_r(\mathcal{D}) = s) =$$

$$\Pr_r(J_r = \perp \wedge S_r(\mathcal{D}) = s) + \sum_{j \in \mathcal{D}} \Pr_r(J_r = j \wedge S_r(\mathcal{D}_{-j}) = s). \tag{13}$$

Next, the summands over $j \in \mathcal{D}$ are decomposed via conditional probabilities. Specifically, the right hand side of Equation (13) equals:

$$\Pr_r(J_r = \perp \wedge S_r(\mathcal{D}) = s) + \sum_{j \in \mathcal{D}} \Pr_r(J_r = j \mid S_r(\mathcal{D}_{-j}) = s) \Pr_r(S_r(\mathcal{D}_{-j}) = s). \tag{14}$$

For any hash-based, order-invariant sketch, the distribution of $S(\mathcal{D})$ depends only on the number of distinct elements in $\mathcal{D}$, and hence the factor $\Pr_r(S_r(\mathcal{D}_{-j}) = s)$ appearing in the $j$th summand of Equation (14) equals $\Pr_r(S_r(\mathcal{D}_{-i}) = s)$, where $i$ is the element of $\mathcal{D}$ referred to in the statement of the lemma. Accordingly, we can rewrite Expression (14) as:

$$\Pr_r(J_r = \perp \wedge S_r(\mathcal{D}) = s) + \sum_{j \in \mathcal{D}} \Pr_r(J_r = j \mid S_r(\mathcal{D}_{-j}) = s) \Pr_r(S_r(\mathcal{D}_{-i}) = s). \tag{15}$$

Clearly, $J_r \neq \perp$ whenever the number of items in the data set, $n$, exceeds $K_r$. Hence, if $n > \sup_r K_r$, $\Pr_r(J_r = \perp \wedge S_r(\mathcal{D}) = s) = 0$. We obtain Equation (4) as desired, namely:

$$\frac{\Pr_r(S_r(\mathcal{D}) = s)}{\Pr_r(S_r(\mathcal{D}_{-i}) = s)} = \sum_{j \in \mathcal{D}} \Pr_r(J_r = j \mid S_r(\mathcal{D}_{-j}) = s).$$

$\square$

We subsequently prove Lemma 2.2 which bounds the sum $\sum_{j \in \mathcal{D}} \Pr_r(J_r = j \mid S_r(\mathcal{D}_{-j}) = s)$ in terms of two parameters that we need to control. The first is related to the "sampling probability" of the sketch, $\pi(s)$ and the second is an unwieldy expectation. Although the expectation may be difficult to compute, we will later show a more practical variant that will be easier for us to leverage algorithmically. For clarity, Lemma A.2 is a restatement of Lemma 2.2 from the main body.

**Lemma A.2.** *Under the same assumptions as Lemmas 2.1 and A.1,*

$$\sum_{j \in \mathcal{D}} \Pr_r(J_r = j \mid S_r(\mathcal{D}_{-j}) = s) = (1 - \pi(s)) \mathbb{E}_r \left( 1 + \frac{K_r}{n - K_r + 1} \bigg| S_r(\mathcal{D}_{-1}) = s \right). \tag{16}$$

**Proof of Lemma 2.2.** We begin by writing

$$\Pr_r(J_r = j \mid S_r(\mathcal{D}_{-j}) = s) =$$

$$\sum_{k=1}^{n} \Pr_r(J_r = j \mid K_r = k, S_r(\mathcal{D}_{-j}) = s) \Pr_r(K_r = k | S_r(\mathcal{D}_{-j}) = s). \qquad (17)$$

To analyze Expression (17), we first focus on the $\Pr_r(J_r = j \mid K_r = k, S_r(\mathcal{D}_{-j}) = s)$ term. Given that $K_r = k$ and $S_r(\mathcal{D}_{-j}) = s$, we know that $J_r = j$ if and only if the items $\{1, \ldots, j-1\}$ are *all* in $\mathcal{K}_r$ *and* item $j$ is not in $\mathcal{K}_r$. The first condition occurs with probability $\frac{\binom{n-(j-1)}{k-(j-1)}}{\binom{n}{k}} = \frac{\binom{k}{j-1}}{\binom{n}{j-1}}$. This is because there are $\binom{n-(j-1)}{k-(j-1)}$ subsets $K_r$ of $\{1, \ldots, n\}$ of size $k$ that contain items $1, \ldots, j-1$, out of $\binom{n}{k}$ subsets of $\mathcal{K}_r$ of size $k$. Meanwhile, $j \notin \mathcal{K}_r$ means that $S_r(\mathcal{D}_{-j}) = S_r(\mathcal{D})$, which occurs with probability that is exactly the complement of the sampling probability $\pi(s)$ (see Equation (3)).

By the above reasoning, the left hand side of Expression (16) equals:

$$\sum_{j \in \mathcal{D}} \sum_{k=1}^{n} \Pr_r(J_r = j \mid K_r = k, S_r(\mathcal{D}_{-j}) = s) \Pr_r(K_r = k | S_r(\mathcal{D}_{-j}) = s)$$

$$= \sum_{k=1}^{n} \sum_{j \in \mathcal{D}} \frac{\binom{k}{j-1}}{\binom{n}{j-1}} (1 - \pi(s)) \Pr_r(K_r = k | S_r(\mathcal{D}_{-1}) = s)$$

$$= (1 - \pi(s)) \sum_{k=1}^{n} \left( 1 + \frac{k}{n-k+1} \right) \Pr_r(K_r = k | S_r(\mathcal{D}_{-1}) = s)$$

$$= (1 - \pi(s)) \mathbb{E}_r \left( 1 + \frac{K_r}{n - K_r + 1} \Big| S_r(\mathcal{D}_{-1}) = s \right). \qquad (18)$$

$\square$

By comparing the LHS of (4) to the right hand side of (18), overall, Lemmas 2.1 and 2.2 show:

$$\frac{\Pr_r(S_r(\mathcal{D}) = s)}{\Pr_r(S_r(\mathcal{D}_{-i}) = s)} = (1 - \pi(s)) \mathbb{E}_r \left( 1 + \frac{K_r}{n - K_r + 1} \Big| S_r(\mathcal{D}_{-1}) = s \right).$$

Hence, $\epsilon$-DP is a consequence of ensuring

$$(1 - \pi(s)) \mathbb{E}_r \left( 1 + \frac{K_r}{n - K_r + 1} \Big| S_r(\mathcal{D}_{-1}) = s \right) \in [e^{-\epsilon}, e^{\epsilon}]$$

and Corollary 2.3 presents the conditions under which this is true.

**Proof of Corollary 2.3.** Since $1 - \pi(s) \leq 1$ for all $s$ and $\frac{n+1}{n - K_r + 1} = 1 + \frac{K_r}{n - K_r + 1} \geq 1$, it follows from Lemma 2.2, (6) and (7) that the right hand side of Equation (4) lies in the interval $[e^{-\epsilon}, e^{\epsilon}]$, as required for an $\epsilon$-DP guarantee.

For the necessity of Condition 6, note that if the universe of possible items is infinite, then for any possible sketch state $s$, there exists an arbitrarily long sequence of distinct items that results in state $s$ if $\pi(s) < 1$. One simply needs to search for a sequence of items which do not change the sketch. Combining Lemma 2.1 with Equation (16) in Lemma 2.2 therefore implies that

$$e^{-\epsilon} < \inf_s (1 - \pi(s)) \qquad (19)$$

and hence $\sup_s \pi(s) < 1 - e^{-\epsilon}$ as claimed. $\square$

As previously described, because the expected value in Lemma 2.2 depends on the unknown cardinality $n$, it is difficult to use. However, we know from Lemma A.1 that there are only a bounded number

of items that actually change the sketch. Thus, we introduce a sketch specific parameter $k_{max} \geq K_r$ which is a bound on the maximum number of items that can change the sketch. Although we trivially have $k_{max} \leq b$ from Lemma A.1, Section 4 in fact shows that $k_max = k$, the number of "buckets" used in the sketch for many popular algorithms.

**Proof of Theorem 2.4.** We can upper bound the expectation on the right hand side of Condition (7) using $k_{max}$ and $n_0$. Corollary 2.3 and solving for $n_0$ then gives the desired result. Specifically, by Corollary 2.3, the sketch satisfies $\epsilon$-DP if:

$$e^\epsilon > \sup_{n \geq n_0} \mathbb{E}_r \left( 1 + \frac{K_r}{n - K_r + 1} \middle| S_r(\mathcal{D}_{-1}) = s \right) \tag{20}$$

$$\geq \sup_{n \geq n_0} \left( 1 + \frac{k_{max}}{n - k_{max} + 1} \right) \tag{21}$$

$$= 1 + \frac{k_{max}}{n_0 - k_{max} + 1}. \tag{22}$$

This is satisfied if $n_0 - k_{max} + 1 > \frac{k_{max}}{e^\epsilon - 1}$ so that

$$n_0 > k_{max} \left( 1 + \frac{1}{e^\epsilon - 1} \right) - 1 = \frac{k_{max}}{1 - e^{-\epsilon}} - 1 \tag{23}$$

$\square$

In summary, all of the technical results leading to Theorem 2.4 are used to show that, provided $\sup_s \pi(s) < 1 - e^{-\epsilon}$ and at least $n_0$ items appear in the stream, then any sketching algorithm from the hash-based order-invariant class will satisfy $\epsilon$-DP. Theorem 2.5 concludes the subsection and shows that if $\pi(s)$ is too large, then there is a small, $\delta$ probability that the sketch is privacy violating. In other words, if there are at least $n_0$ items in the stream, but the hash-based downsampling rate $\pi(s)$ is not small enough, then there is a tiny $\delta$ chance the sketch may not be $\epsilon$-DP, and hence the sketch in this case is $(\epsilon, \delta)$-DP.

### A.3    Algorithmic Approach: Sections 2.1 and 2.2

We separate our algorithms into three regimes described by Algorithms 1a, 1b, 1c. Algorithm 1a is a "base" sketching algorithm chosen from the hash-based, order-invariant class, for example, a HyperLogLog or Bottom-$k$ sketch. No modifications are made to the inner workings of the algorithm but it must be implemented using a perfectly random hash function (see the final paragraph of Section 1.1). We show in Theorem 2.6 that on streams with at least $n_0$ items that the quality of the estimator is related to the downsampling probability as presented in Theorem 2.5. It is used to relate the probability of a privacy-violation back to the estimate quality, rather than simply the state of the sketch.

**Proof of Theorem 2.6.** As $\tilde{N}$ is invertible and decreasing, $P(\hat{N}(S_r(\mathcal{D})) < \tilde{N}(\pi_0)) = P(\pi(S_r(\mathcal{D})) > \pi_0) = \delta.$ $\square$

The final result of Section 2, Corollary 2.7, shows that by modifying Algorithm 1a, we can strictly enforce Conditions 8 and 9 to guarantee $\epsilon$-DP. Namely, by appropriately downsampling using $\pi_0 = 1 - e^{-\epsilon}$, Algorithm 1b is $\epsilon$-DP if we have an a prior guarantee that the number of items in the stream is at least $n_0$. If this guarantee is not known in advance, then the same $\pi_0$ is used for sampling, alongside the insertion of "phantom" elements to satisfy the minimum cardinality condition.

**Proof of Corollary 2.7.** Under their respective assumptions, Algorithms 1b and 1c, `DPSketchLargeSet` and `DPSketchAnySet` respectively, satisfy Conditions (8) and (9) of Theorem 2.4. $\square$

## A.4 Private Sketches via Merging: Section 2.3

Algorithm 3 converts a non-private sketch, $S_r$ into an $\epsilon$-DP sketch by merging it with a noisy sketch $T$. Merging requires the same seed to be used so we suppress this notation in the subsequent writing. The merge step is a property of the specific sketching algorithm used and operates on the sketch states $s$ and $s'$ so we also overload the notation to denote the merge over states by $s \cup s'$.

Since sketch $s$ is already constructed, items cannot be first downsampled in the building phase the way they are in Algorithms 1b and 1c. To achieve the stated privacy, Algorithm 3 constructs a noisily initialized sketch, $t$, which satisfies both the downsampling condition (Condition (8)) and the minimum stream cardinality requirement (Condition (9)) and returns the merged sketch $s \cup t$. Hence, the sketch will satisfy both conditions for $\epsilon$-DP, as shown in Corollary A.3

**Corollary A.3.** *Regardless of the sketch $s$ provided as input to the function* `MakeDP` *(Algorithm 3),* `MakeDP` *yields an $\epsilon$-DP distinct counting sketch.*

*Proof.* Given sketches $S, T$ with states $s$ and $t$, respectively, we claim that any item that does not modify $T$ also cannot modify the merged sketch $S \cup T$ by the order-invariance of $S, T$. To see this, let $\mathcal{D}_S$ and $\mathcal{D}_T$ respectively denote the streams that were processed by sketches $S$ and $T$, and consider an item $i$ that does not appear in $\mathcal{D}_S$ or $\mathcal{D}_T$ and whose insertion into $\mathcal{D}_T$ would not change the sketch $T$. Since the state of the sketch $T$ is the same after processing $\mathcal{D}_T \circ i$ as it was after processing $\mathcal{D}_T$, $S \cup T$ is also the sketch of $\mathcal{D}_T \circ i \circ \mathcal{D}_S$, where $\circ$ denotes stream concatenation. By order-invariance, $S \cup T$ is also a sketch for $\mathcal{D}_T \circ \mathcal{D}_S \circ i$. Also by order-invariance, $S \cup T$ is a sketch for $\mathcal{D}_T \circ \mathcal{D}_S$. Hence, we have shown that the insertion of $i$ into $\mathcal{D}_T \circ \mathcal{D}_S$ does not change the resulting sketch.

It follows that $\pi(s \cup t) \le \pi(t) \le \pi_0$, where the last inequality holds by the stopping condition of the loop in `DPInitSketchForMerge` (Algorithm 2b). Hence, `MakeDP` also satisfies Condition (8). The requirement that $v \ge n_0$ in `DPInitSketchForMerge` also ensures that $S \cup T$ is a sketch of a stream satisfying Condition (9). Hence, Theorem 2.4 implies that the sketch $S \cup T$ returned by `MakeDP` satisfies $\epsilon$-DP. Since the additional value $v$ that affects the estimate returned by `MakeDP` does not depend on the data, there is no additional privacy loss incurred by returning it. □

## B Utility Proofs: Section 3

in this section we present the proofs showing that differentially private sketches have the same asymptotic performance as non-private sketches. Namely, they remain unbiased and have the same variance as the number of unique items in the stream grows. These results apply to Algorithms 1a-1c and Algorithm 3. First we will show unbiasedness.

**Proof of Theorem 3.1.** Trivially, Algorithm 1a is unbiased by assumption, as it does not modify the base sketch. Given $V$, there are $Z \sim Binomial(n + V, p)$ items added to the base sketch. Since the base sketch's estimator is unbiased, $\mathbb{E}(\hat{N}_{base}|Z) = Z$. Algorithms 1b, 1c, and Algorithm 3 all have expectation:

$$\mathbb{E}\left(\hat{N}(S_r(\mathcal{D}))|V\right) = \mathbb{E}\left(\mathbb{E}\left(\frac{\hat{N}_{base}}{p} - V \middle| V, Z\right)\right)$$

$$= \mathbb{E}\left(\frac{Z}{p} - V \middle| V\right) = n + V - V = n.$$

□

Next we establish the variance properties of the sketching algorithms. This involves expressing the variance of estimates from the algorithms in terms of the "base sketch" estimator.

**Corollary B.1.** *The variance of the estimates produced by Algorithms 1a-1c and 3 is given by*

$$\mathsf{Var}\left(\hat{N}(S_r(\mathcal{D}))\right) = \mathbb{E}\left(\mathsf{Var}\left(\frac{\hat{N}_{base}}{p}\middle|V\right)\right). \tag{24}$$

*Proof.* This follows from the law of total variance and the fact that the estimators are unbiased. □

Finally, we can leverage Corollary B.1 to show there is no asymptotic increase in variance caused by the extra steps added to make a sketch private.

**Proof of Theorem 3.2.** Let $Z \sim Binomial(n + v, \pi_0)$ denote the actual number of items inserted into the base sketch. From Corollary B.1 and since $V$ is constant, the variance is

$$\text{Var } \hat{N}(S_r(\mathcal{D})) = \left( \text{Var} \left( \left. \frac{\hat{N}_{base}}{\pi_0} \right| V = v \right) \right)$$

$$= \left( \frac{\mathbb{E} \text{Var}(\hat{N}_{base}|Z) + \text{Var } \mathbb{E}(\hat{N}_{base}|Z)}{\pi_0^2} \right)$$

$$\leq \left( \frac{\mathbb{E} Z^2/c + \text{Var}(Z)}{\pi_0^2} \right)$$

$$= \frac{(\mathbb{E} Z)^2}{c\pi_0^2} + \frac{\text{Var}(Z)(c + 1)}{c\,\pi_0^2}$$

$$= \frac{(n + v)^2}{c} + \frac{(n + v)(1 - \pi_0)}{c\pi_0}.$$

Trivially, $\frac{\text{Var}_n(\hat{N})}{\text{Var}_n(\hat{N}_{base,n})} = \frac{(n+v)^2}{n^2} + O(1/n) \to 1$ as $n \to \infty$. □

**Corollary B.2.** *Assume that the conditions of Theorem 3.2 hold. Further assume the base sketching algorithm satisfies an $(\epsilon, \delta_n)$ privacy guarantee where $\delta_n \to 0$ as $n \to \infty$. For any given $n^* > n_0$, we say Algorithm 3 succeeded if $V < n^*$. Then with probability at least $1 - \delta_{n^*}$*

$$\text{Var}_n(\hat{N}|Success) \leq \frac{(n + n^*)^2}{c} + \frac{(n + n^*)(n_0 + \pi_0^{-1})}{k_{max}}$$

*and*

$$\frac{\text{Var}_n(\hat{N}|V)}{\text{Var}_n(\hat{N}_{base,n})} \xrightarrow{p} 1 \quad as \ n \to \infty,$$

*where $X_n \xrightarrow{p} 1$ denotes convergence in probability: $P(|X_n - 1| < \Delta) \to 1$ as $n \to \infty$ for any $\Delta > 0$.*

## C  Concrete Examples: Section 4

In this section we provide the proofs for results showing that popular sketches are $(\epsilon, \delta)$-DP. We also provide further discussion for Adaptive Sampling that is omitted from the main body.

### C.1  FM85

**Proof of Theorem 4.1.** To obtain an $(\epsilon, \delta)$ guarantee, note that bit $s_{ij}$ in the sketch has probability $2^{-i}/k$ of being selected by any item. If $v = \lceil -\log_2 \pi_0 \rceil$ and all bits $s_{ij}$ with $j \leq v$ are flipped, then $\pi(s) < \pi_0$. The probability $\text{Pr}_r(s_{ij} = 0) = (1 - 2^{-i}/k)^n \leq \exp(-2^{-i}n/k)$. A union bound gives that $\text{Pr}_r(\pi(S_r(\mathcal{D})) \geq \pi_0) \leq k \sum_{i=1}^v \exp(-2^{-i}n/k) \leq kv \exp(-2^{-v}n/k) = kv \exp\left(-\tilde{\pi}_0 \frac{n}{k}\right)$ where $\tilde{\pi}_0 = 2^{-v} \leq \pi_0$. □

Recall that the quantity $k_{max}$ for FM85 is larger than all other sketches by either $32$ or $64$, the number of bits used in the hash function. Thus, FM85 requires a larger minimum number of items in the sketch to guarantee privacy, see Equation (9). However, the sketch is highly compressible as, for large $n$, each bitmap has entropy of approximately $4.7$ bits [16]. More recent works have placed this numerical result on firmer theoretical footing [19], and in fact shown that the resulting space-vs.-error tradeoff is essentially *optimal* amongst a large class of sketching algorithms. A

practical implementation of the compressed sketch can be found in the Apache DataSketches library [1].[5] It achieves close to constant update time by buffering stream elements and only decompressing the sketch when the buffer is full.

Our results imply the above compressed sketches can yield a relaxed $(\epsilon, \delta_n)$-differential privacy guarantee when the number of inserted items is $n \geq n_0$ (Equation (23)). If the size of the sketch in bits is $b$, the sketch is $\epsilon$-differentially private if $n > \frac{b}{1-\exp(-\epsilon)}$ or equivalently $b < n(1 - \exp(-\epsilon))$. Thus, $\delta_n = \Pr_r(b \geq n(1 - \exp(-\epsilon))$.

## C.2 Linear Probabilistic Counting

**Proof of Theorem 4.2.** $S_r(D)$ is not privacy violating (i.e., $\pi(s) < \pi_0$) if $\pi(S_r(\mathcal{D})) = p(1 - B/k) < \pi_0$ or, equivalently, $B > k(1 - \pi_0/p)$. Note that $G_i \sim Geometric(p(1 - i/k))$ items must be added for the number of filled bits to go from $i$ to $i + 1$.

We can use a tail bound for the sum of geometric random variables [15]. Assume that $n \geq \frac{k-1}{1-\exp(-\epsilon)} \geq n_0$ so that Condition 9 is satisfied. If $n > \mu_0$ then

$$\delta \leq P\left(\sum_{i=0}^{b_0} G_i > n\right) \tag{25}$$

$$\leq \exp\left(-\tilde{\pi}_0(n/\mu_0 - 1 - \log(n/\mu_0))\right) \tag{26}$$

$$= \frac{\mu_0}{n} \exp\left(-\tilde{\pi}_0(n/\mu_0 - 1)\right). \tag{27}$$

$$\tag{28}$$

The number of expected items needed to fill $b_0$ bits if $b_0 \geq 1$ is

$$\mu_0 := \sum_{i=0}^{b_0-1} \frac{1}{p} \frac{1}{1-i/k} \tag{29}$$

$$< \frac{k}{p}\left(\log\left(\frac{k}{k-b_0}\right) + 1/(2k) - 1/(2(k-b_0)) + \frac{1}{12(k-b_0)^2}\right) \tag{30}$$

$$= \frac{k}{p}\log\left(\frac{k}{k-b_0}\right) - \frac{1}{p}\frac{b_0}{2(k-b_0)} + \frac{1}{p}\frac{k}{12(k-b_0)^2} \tag{31}$$

$$= \frac{k}{p}\log\left(\frac{k}{k-b_0}\right) - \frac{1}{p}\frac{6b_0(k-b_0)-k}{12(k-b_0)^2} \tag{32}$$

$$< \frac{k}{p}\log\left(\frac{k}{k-b_0}\right). \tag{33}$$

$\square$

## C.3 Bottom-$k$ / KMV

**Proof of Theorem 4.3.** The value $\pi(s)$ is equal to the $k^{th}$ minimum value. If $X > k$, then the $k^{th}$ minimum value is $< \pi_0$ and Condition 8 is satisfied. Thus, $\delta$ is the failure probability. The bound follows directly from Bernstein's inequality:

$$P(X \leq k) = P(n - X > n - k) = P((n - X) - n(1 - \pi_0) > n\pi_0 - k) \tag{34}$$

$$\leq \frac{1}{2}\frac{(n\pi_0 - k)^2}{n\pi_0(1 - \pi_0) + 1/3} \tag{35}$$

$$= \frac{1}{2}\frac{(\pi_0 - k/n)^2}{\pi_0(1 - \pi_0) + 1/3n^2}n \tag{36}$$

$\square$

---

[5]It is referred to as CPC, short for compressed probabilistic counting.

Table 2: Comparison of the utility bounds for related work. The only corresponding result to ours in [5] is Theorem 4.2, which shows that the LogLog sketch is $(\epsilon, \delta')$-DP provided $n \geq n_0'$ for $n_0'$ at least a factor 8 larger than our $n_0$ from (23). Our approaches, Algorithms 1a-1c simultaneously achieve the best utility and update time,as well as a tighter privacy bound.

| Algorithm | Privacy | Utility (Relative Error): $\gamma$ | Update Time |
|---|---|---|---|
| 1a | $(\epsilon, \delta)$ for $n \geq n_0$ | $\frac{1.04}{\sqrt{k}}$ | $O(1)$ |
| 1b, 1c | $\epsilon$-DP for $n \geq n_0, 1$, respectively | $\frac{1.04}{\sqrt{k}}$ | $O(1)$ |
| [22] | $(\epsilon, 0)$ and $(\epsilon, \delta)$-DP | $\frac{10 \ln(1/\beta)^{1/4}}{\sqrt{k}}$ | $O(k)$ |
| [5] | $(\epsilon, \delta)$ for $n \geq n_0' \geq 8n_0$ | $\frac{1.3}{\sqrt{k}}$ | $O(1)$ |

## C.4 Adaptive Sampling

Wegman's adaptive sampling is similar to the bottom-$k$ sketch but does not require the sketch to store exactly $k$ hashes. Instead, it maintains a threshold $p$ and stores all hash values less than $p$. Whenever the sketch size exceeds $k$, then the threshold is cut in half and only values under the threshold are retained. This ensures that processing a stream of length $n$ takes expected $O(n)$ time rather than $O(n \log k)$ as in a max-heap-based implementation of Bottom-$k$.

It is order invariant since the sketch only depends on the number of hash values under each of the potential thresholds and not the insertion order. Since at most $k$ hashes are stored, and the sketch is modified only if one of these hashes is removed, like KMV a.k.a. Bottom-$k$, the maximum number of items that can modify the sketch by removal is $k_{max} = k$. In Corollary 4.4 we showed that Adaptive Sampling with $k$ buckets has the same privacy behavior as a Bottom-$k$ sketch.

**Proof of Corollary 4.4.** Consider sketches $S_r^{AT}(\mathcal{D})$, $S_r^{KMV}(\mathcal{D})$ using the same hash function. Since the threshold in adaptive sampling is at most the $k^{th}$ minimum value, $\pi(S_r^{AT}(\mathcal{D})) \leq \pi(S_r^{KMV}(\mathcal{D}))$. So $\pi(S_r^{KMV}(\mathcal{D})) < \pi_0 \implies \pi(S_r^{AT}(\mathcal{D})) < \pi_0$. □

## C.5 HyperLogLog

**Proof of Theorem 4.5.** In HLL, the sampling probability $\pi(s) = k^{-1} \sum_{i=1}^{k} 2^{-s_i}$ here $s_i$ is the value in each bin. Thus, if all bins have value $s_i > -\log_2 \pi_0$, then $\pi(s) < \pi_0$. Let $C_i$ be the event that $s_i > -\log_2 \pi_0$. Then $P(\neg C_i | n) \leq (1 - \pi_0/k)^n \leq \exp(-\pi_0 n/k)$. A union bound gives $\Pr_r(\pi(S_r(\mathcal{D})) \geq \pi_0) \leq k \exp(-\pi_0 n/k)$. □

# D Further Empirical Details

**Experiment 1: Update Time.** We implement the regular, non-private HLL using a 32-bit non-cryptographic MurmurHash. Our Private HLL (PHLL) is implemented in the Algorithm 1b setting with a 256-bit cryptographic hash function, SHA-256. In this model, PHLL employs the same algorithm as HLL but uses an extra downsampling step and rescales the estimator. QLL, the $\epsilon$-DP algorithm of [22] also uses SHA-256. We record the time to populate every sketch with $2^{10}$ updates with $k \in \{2^7, 2^8, \ldots 2^{12}\}$ buckets. Each marker represents the mean update time over all updates and the curves are the evaluated mean update time over 10 trials.

We only implement [22] with $\gamma = 1.0$ because the running time is independent of $\gamma$. Since the running time is independent of the privacy parameter for all methods, we test the total privacy budget $\epsilon = 1.0$. All methods are implemented in Python so the speed performance could be optimized using a lower-level implementation. In absolute terms, we find that for $2^{10}$ updates, HLL needs $8 \times 10^{-6}$ seconds compared to $9 \times 10^{-6}$ seconds for PHLL. Both methods have a standard deviation of about $5 \times 10^{-7}$. On the other hand, [22] needs between $2 \times 10^{-4}$ and $5 \times 10^{-3}$ seconds with a standard deviation on the order of $10^{-7}$ which is imperceptible on the scale of Figure 1a.

**Experiment 2: Utility-space tradeoff (Figure 3).** We simulate each of the algorithms (HLL, PHLL, QLL) on an input stream of cardinality $n = 2^{20}$ over 100 independent trials. The privacy budget is

fixed at $\epsilon = \ln(2)$ meaning $\pi_0 = 1/2$ (6). The number of buckets was varied in $k \in \{2^7, 2^8, \dots 2^{12}\}$ and we record the simulated **total sketch size** $:= k \cdot \log_2 \max_i s_i$, which is the space for the largest bin value $s_i$ for a sketch with $k$ buckets. In this example, $k_{max} = k$ for HLL (Table 1) so that (9) $n_0 = 2(k-1)$. As $n \geq n_0$ PHLL is $\epsilon$-DP in this setting by Theorem 2.4. Essentially, PHLL under Algorithm 1b employs HLL as the base sketch with a stream downsampled by $\pi_0 = 1/2$.

For QLL, the relative error $\gamma_{QLL} = \frac{10 \ln(1/\beta)^{0.25}}{\sqrt{k}}$ [22, Theorem 2.6] depends on the sketching failure probability $\beta$ and the number of buckets. This is a factor $10 \ln(1/\beta)^{0.25}/1.04$ worse than our bound for $\gamma_{PHLL}$ and occurs with probability at least $1 - \beta$. We set $\beta = 0.05$ so that $\gamma_{QLL} \approx 7.49/\sqrt{k}$ for [22], while our method has $\gamma_{PHLL} = 1.04/\sqrt{k}$. Only the base-$(1 + \gamma_{QLL})$ harmonic estimator was tested for closest comparison to HLL and our work.

Figure 3 plots the utility in relative error against the total space usage of the methods. We see that PHLL is indistinguishable from HLL. The utility of QLL appears comparable to (P)HLL and is better than its worst-case relative error guarantees, yet this comes at the cost of using more space than our sketch, PHLL. In absolute terms PHLL consumes approximately 90% of the space used by QLL when $k = 2^7$, which decreases to roughly 65% when $k = 2^{12}$. The fractional reduction in space usage of PHLL over QLL is because $\gamma$ decreases as $k$ grows and reducing $\gamma$ affects the $Geometric(\frac{\gamma}{1+\gamma})$ hash function used to select the bin values in QLL; smaller $\gamma$ result in larger bin values which inflate the size of the QLL sketch. In fact, we find that the mean total space usage of QLL compared to PHLL, is larger by a factor of $O(\log k)$. This agrees with the ratio of theoretical space bounds of [22] and our work, as illustrated in Figure 1b.

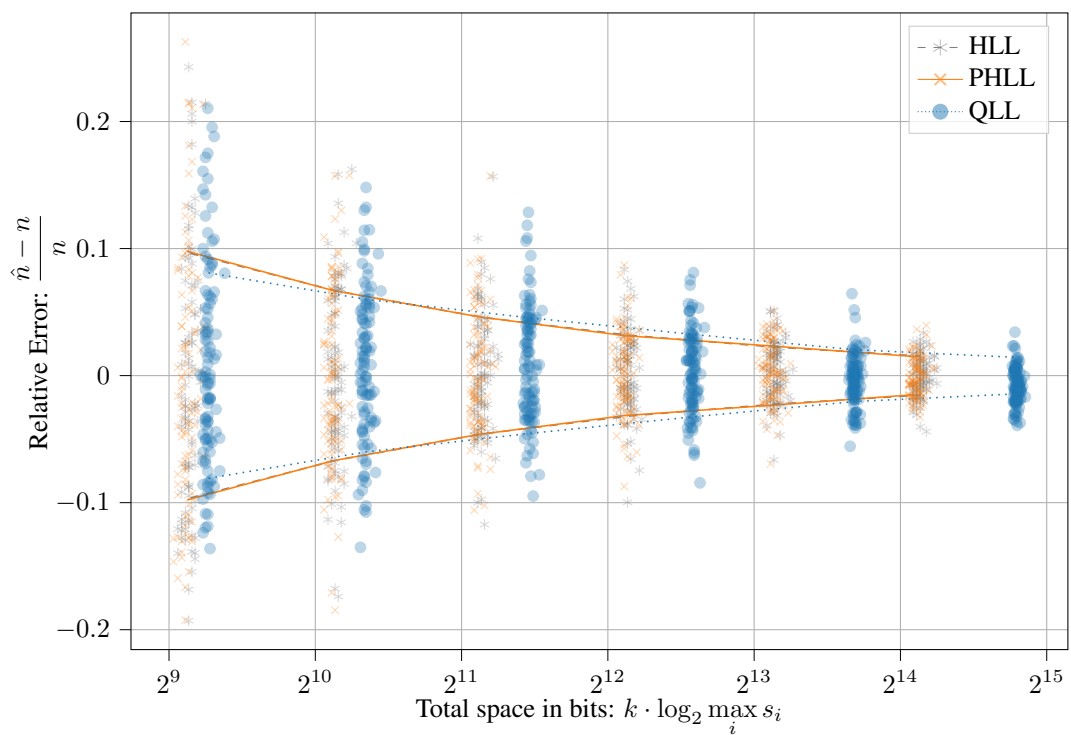

Figure 3: The curves represent empirical standard deviations of the estimates. For our method, PHLL, this matches the error bound $1.04/\sqrt{k}$ as indicated from Table 1. Empirically, QLL has a nearly matching standard deviation to PHLL, despite a suboptimal utility bound as seen in Table 2.