# OpenReview forum: "Order-Invariant Cardinality Estimators Are Differentially Private"
_NeurIPS.cc/2022/Conference — NeurIPS 2022 Accept_

### Official Review · Reviewer_8qSU · 2022-06-26

**Rating:** 6
**Confidence:** 5
**Soundness:** 3 good
**Presentation:** 3 good
**Contribution:** 3 good

**Summary:**

The authors prove that nearly all **hash-based**, **order-invariant** cardinality estimators are differentially private once combined with a down-sampling procedure. This class includes several well-celebrated cardinality streaming algorithms such as Flajolet-Martin, HyperLogLog, and Bottom-k.

**Questions:**

Please address the below questions in the rebuttal.
1. Give an empirical comparison of accuracy between QLL and PHLL. Discuss the reason for different trade-offs between space, update time and accuracy.
2. Change the title to avoid overclaim.
3. Discuss the relevant works mentioned above in Related Work section.
4. Fix line 135.
5. (Optional) Discuss whether the down-sampling procedure can be used for privacy amplification.

**Limitations:**

The authors should exhibit accuracy evaluation results and discuss the limitations if it is worse than QLL.

**Strengths And Weaknesses:**

I really enjoyed reading the paper. The text is well-organized and the authors did a good job explaining the tedious math proof in an intuitive and easy-to-follow way. Overall I think the paper makes solid contribution and passes the bar of acceptance. Concrete strengths and weaknesses are listed below and I am willing to further improve the score if the authors can address most of the weaknesses in their rebuttal.

**Strength**:
1. The paper considers a class of cardinality estimators and proves DP for them as long as they satisfy several conditions, compared to prior works only considering one specific sketching algorithm.
2. The paper evaluates the update time and space ratio which exhibits the superiority of the proposed algorithm.

**Weakness**:
1. I am a little bit worried about the accuracy of the proposed sketching with down-subsampling. In Section 5, only update time and space ratio are evaluated. However, it is unclear how QLL and PHLL compares in terms of accuracy. If QLL is much better in accuracy, then a more thorough discussion of the different trade-offs between QLL and PHLL is needed.
2. Subsampling is also known to provide privacy amplification for DP guarantees. In the proposed algorithm, down-sampling is used to eliminate the $\delta$ term. Does it also contribution to privacy amplification?
3. The related work section ignores some recent progress on DP streaming algorithms. These works consider a broader class of streaming algorithms in which cardinality estimation is a special case of order 0. Specifically [1]-[3] consider another special case of order 2 and [5] considers cases of real orders from 0 (exclusive) to 1.

[1] "The johnson-lindenstrauss transform itself preserves differential privacy." Jeremiah Blocki, Avrim Blum, Anupam Datta, and Or Sheffet. FOCS 2012.

[2] "Randomness Efficient Fast-Johnson-Lindenstrauss Transform with Applications in Differential Privacy and Compressed Sensing." Jalaj Upadhyay.  CoRR, abs/1410.2470, 2014.

[3] "Differentially private ordinary least squares." Or Sheffet. ICML 2017.

[4] "Differentially Private Fractional Frequency Moments Estimation with Polylogarithmic Space." Lun Wang, Iosif Pinelis, and Dawn Song. ICLR 2022.

4. The title needs change. Maybe "Nearly all hash-based order-invariant cardinality estimators are differentially private"? The current one seems to be an overclaim.
5. Line 135: The sentence "By further conditioning on the total number of items $K_r$ that can be removed without changing the sketch" seems misleading. $K_r$ is the total number of items that change the sketch once removed if I understand correctly according to Equation (2)?

---

> ### Author Response · Authors · 2022-08-01
> **Response to Reviewer 8qSU**
>
> - I am a little bit worried about the accuracy of the proposed sketching with down-subsampling. In Section 5, only update time and space ratio are evaluated. However, it is unclear how QLL and PHLL compare in terms of accuracy.  If QLL is much better in accuracy, then a more thorough discussion of the different trade-offs between QLL and PHLL is needed.
>
> We agree that we should better indicate the accuracy in the plots, and propose to plot space vs error (rather than buckets) in figure 1b. We note that we point to figure 3 in the supplement to show that there is little difference in error between QLL and our modification of HLL for a given number of bins.
>
> We also note that QLL implicitly performs hash-based downsampling by imposing a minimum bin value in the sketch; please see the definition of $\alpha_{min}$ in Line 1 of Algorithm 2 from [20]. Indeed, if we treated QLL as $k$ independent sketches with 1 bin each rather than a single sketch with $k$ bins, our analysis essentially recovers the QLL privacy analysis but expresses it in slightly different terms.
>
> - Subsampling is also known to provide privacy amplification for DP guarantees. In the proposed algorithm, down-sampling is used to eliminate the delta term. Does it also contribution to privacy amplification?
>
> Yes.  The downsampling rate is related to the privacy through Equation (6) and more downsampling improves privacy.
>
> - The related work section ignores some recent progress on DP streaming algorithms.
>
> We will add discussion of these related works. They use quite different techniques to prove DP results for other types of $L_p$ functions over data streams, but not $L_0$.
>
> - The title needs change. Maybe "Nearly all hash-based order-invariant cardinality estimators are differentially private"? The current one seems to be an overclaim.
>
> On reflection, we can see why the current title may cause confusion. Our intention was to communicate that all algorithms used in practice that we are aware of are hash-based and order-invariant, and hence our results apply to them. We will alter the title to mitigate the concern of an overclaim.
>
> - Line 135: The sentence "By further conditioning on the total number of items that can be removed without
> changing the sketch" seems misleading. Is the total number of items that change the sketch once removed if I understand correctly according to Equation (2)?
>
> We will clarify this point, which should read  "By further conditioning on the total number of items that, when removed, can change the sketch".  The calligraphic $\cal{K}_r$ in (2) is the _set_ of items that change the sketch on removal, meanwhile the
> non-calligraphic $K_r$ is the size of this set.

---

### Official Review · Reviewer_wzWL · 2022-06-29

**Rating:** 7
**Confidence:** 4
**Soundness:** 4 excellent
**Presentation:** 3 good
**Contribution:** 4 excellent

**Summary:**

The paper studies the general problem of cardinality estimators in insertion only streams with differential privacy. The authors present a novel and general analysis framework that establishes the privacy property of a vast set of algorithms. The framework shows that any algorithm that is order invariant, based on a purely random hash,  limited in space, and with some sensitivity property is DP.  They show that many standard algorithms respect this property with a simple analysis. This results in improved approximation/privacy tradeoffs. More precisely they show that existing algorithms (not designed for privacy) are private on large enough streams or can be made private with a simple downsampling of the input. This is very useful in practice as it allows to reuse original implementations. The analysis framework seems easy to use and general. The authors also experiment with the algorithms derived.
The results hold a slightly relax setting for DP that involves cryptographic assumptions but this framework seems very reasonable.

**Questions:**

Would it be possible to clarify in a table what novel (or improved) tradeoffs between accuracy and privacy the method allows for the algorithms presented (vs the best known result)?

Do you think the work can be adapted to the continual release model?

Can you add a short statement that the algorithm gets better bound than simply running the non-private algorithm and then perturbing the output?

**Limitations:**

No negative impact to society

**Strengths And Weaknesses:**

+ general framework for the analysis of algorithms
+ results hold for well known algorithm with no or limited modifications to their implementations
- slightly less strong setting for DP (requires cryptographic assumptions on secrecy of hash functions and indistinguishability from random distribution).
- works in the single release model not in the continual release model which is more applicable in stream settings.

---

> ### Author Response · Authors · 2022-08-01
> **Response to reviewer wzWL**
>
> - Would it be possible to clarify in a table what novel (or improved) tradeoffs between accuracy and privacy the method allows for the algorithms presented (vs the best known result)?
>
> We kindly point the reviewer to Table 2 in the supplementary material that outlines the differences between ours and prior work. We can put a note in the main text referring readers to this table.
>
> - Do you think the work can be adapted to the continual release model?
>
> We suspect not. In very large cardinality regimes the private sketch behaves almost the same as the original, base sketch. An attacker that observes which items actually change the sketch perfectly identifies a subset of items in the set. We believe this is an interesting question for future research. We note that, in industry, these types of sketches are often not used in continual release models as they are typically used in regular SQL queries or are kept in some data store and are queried infrequently in comparison to the number of updates to the sketch's state.
>
> - Can you add a short statement that the algorithm gets better bound than simply running the non-private algorithm and then perturbing the output?
>
> Yes, we can add such a statement. We expect that such an algorithm would be less accurate because even the local sensitivity of the underlying non-private sketch is proportional to the true cardinality $n$ with high probability (since the local sensitivity depends on $1/\pi(s)$ (which is typically proportional to the estimate $\hat{n}$ for large $n$). Thus, for high cardinalities, a method that perturbs a non-private algorithm would be expected to have an error that is at least a multiplicative factor greater than our algorithm.

---

> > ### Comment · Reviewer_wzWL · 2022-08-09
> > **Thanks**
> >
> > Thanks for the comments.

---

### Official Review · Reviewer_J5RJ · 2022-07-07

**Rating:** 6
**Confidence:** 4
**Soundness:** 4 excellent
**Presentation:** 4 excellent
**Contribution:** 3 good

**Summary:**

The paper proposes a general technique for proving that hash-based,
order-invariant sketching algorithms for cardinality (which encompasses most
known sketching algorithms for the problem) satisfy DP. The DP parameters depend
on an unconditional upper bound on the probability that a new element causes the state
of the algorithm to change, as well as a condition that the number of unique
elements in the stream be higher than some value. When the sketching algorithm
do not satisfy the conditions, they propose a lightweight, blackbox algorithm
that downsamples the stream and adds in "fake" unique elements, and show this
has no utility drawback and will satisfy privacy.

They derive tighter privacy guarantees for specific streaming algorithms and
validate their privacy and utility results.

**Questions:**

In the introduction, what are the paramters b,k? They should be explained here.
Is the privacy analysis close to being optimal or can a more fine-grained approach be
made, such as not having an unconditional upper bound on the stream update chance
\pi_0 but rather a time-dependent one?

**Limitations:**

    Sketching algorithms by their very nature seem to store a tiny amount of
    data and heavily use randomness. Thus, they should be close to being
    ``private'' already. It would be helpful if the authors could private a toy
    example where blatant non-privacy of the sketching algorithm is
    exposed.

    This also raises the issue that perhaps the privacy analysis and techniques
    only work in this particular sketching setting. It would be
    helpful if the authors could compare and contrast these methods to other
    privacy work in sketching in related works section.

**Strengths And Weaknesses:**

  + The results are general: they are able to make sketching algorithms for
  cardinality satisfy DP

  + The privacy overhead is small in terms of utility cost and computational
  cost

  + Their algorithm is even able to take non-private sketches which are costly
  to recompute and combine them
  with private sketches such that the entire computation will satisfy DP.

  - One possible drawback is that the problem
    intuitively seems slightly ``easy'', see the limitations section.

---

> ### Author Response · Authors · 2022-08-01
> **Response to reviewer J5RJ**
>
> - In the introduction, what are the parameters b,k?
>
> $k$ is the number of buckets in the sketch and $b$ is the space bound in bits.
> We will change the sentences as follows:
> Line 33 revised: “can be k times slower than an HLL sketch that has $k$ buckets”
> Line 68 revised: “And a space bound $b$, measured in bits”.
>
> - Is the privacy analysis close to being optimal or can a more fine-grained approach be made, such as not having an unconditional upper bound on the stream update chance \pi_0  but rather a time-dependent one?
>
> The analysis is optimal in the sense that Equation (5) is the exact likelihood ratio. The resulting sufficient conditions (6) & (7) can be considered asymptotically tight. If $n \to \infty$, (6) is necessary and (7) must hold on a set of sketch states with probability converging to 1 (since 1-\pi(s) \to 1 in probability). However, the expectation in (7) is difficult to compute in general, so we relax the condition to that appearing in Equation (9). For a sketch such as $k$-minimum values, even Equation (9) is tight, because the variable $K_r$ appearing inside the expectation in Equation (7) is fixed to $k$, independent of the algorithm's randomness.
>
> We do not fully understand the second part of the question. However, we note that for epsilon-DP we require the unconditional bounds on $\pi_0$, but for $(\epsilon, \delta)$-DP we avoid the unconditional bound on $\pi_0$ since over time, there are enough updates to ensure $\pi_0$ is small enough.
>
> - It would be helpful if the authors could provide a toy example where blatant non-privacy of the sketching algorithm is exposed.
>
> In a sense, our work shows that no non-trivial examples of blatant non-privacy exist as the sketches are inherently private under the assumption that the hash seed is secret.
> The exception is when the stream is very short (e.g. of length 1). In this case, the base sketches often return an exact or almost exact answer with probability 1. Blatant non-privacy will trivially hold, as exact answers are incompatible with privacy.

---

### Official Review · Reviewer_da9e · 2022-07-11

**Rating:** 7
**Confidence:** 4
**Soundness:** 4 excellent
**Presentation:** 3 good
**Contribution:** 4 excellent

**Summary:**

A sketching algorithm for the distinct elements problem maps a stream of elements from some universe into a small space data structure (the sketch), such that an approximation to the number of distinct elements in the stream can be recovered from the data structure with high probability. Sketches thus compress the data, and it stands to reason that the task of designing a good sketch is aligned with the goal of preserving privacy with respect to the data in the stream. Recent work has confirmed this intuition, and shown that some specific randomized sketches for the distinct elements problem preserve differential privacy, provided that

* the randomness used by the sketch is hidden from the adversary;
* the data stream contains sufficiently many elements: this can be achieved by padding.

The analyses in prior work are sketch-specific, and sometimes require modifying the existing sketch in ways that may affect its practicality.

The present paper unifies this line of work, and, in a way, shows that the privacy preserving properties of distinct element sketches are a deeper property, common to a large class of sketches. Namely, all sketches based on random hashing that are invariant under permuting and duplicating elements satisfy differential privacy, and the privacy parameter depends on two quantities associated with the sketch. One of these quantities is automatically bounded when the stream contains enough elements, and the other is usually small enough with high probability, and, in any case, can be made small enough by subsampling the stream. This unified analysis gives improved bounds, in some cases, and allows ensuring privacy for the most efficient and practical variants of the sketches.

**Questions:**

The explanation on lines 177-184 is not very clear. The last sentence of the paragraph is a little puzzling. Looking at the supplementary material mostly clears this up.

There are some typos in the supplementary material: at least one missing reference, one typo that says “n < n_0” rather than “n > n_0” while referencing (23). Please proofread this part of the paper.

Do you think it’s possible to prove privacy for these sketches if the hash function is public but the stream is preprocessed in some random way?



**Limitations:**

I think limitations are addressed adequately.

**Strengths And Weaknesses:**

The unified analysis in the paper seems to get to the core of why distinct element sketches preserve privacy. The authors have found a good abstraction that allows isolating what makes a sketch private. Thus the proofs are simple, but give strong results for a wide class of algorithms. This makes the paper much more satisfying than papers that analyze just one sketch, even though there are some nice ones in that category as well.

One minor weakness may be that the unity of the results is somewhat undermined by the fact that $\pi(s)$ is not always bounded, which necessitates either a sketch-specific analysis, or preprocessing the stream by subsampling. The fix is simple, however, and does not affect error much. Also, even boiling down the privacy properties of a sketch to just one or two parameters is already quite interesting.

A weakness of this line of work is that the proof of privacy requires the random hash to be secret, which means that the privacy properties do not hold in a distributed setting. I do not hold this against this paper in particular, however.

---

> ### Author Response · Authors · 2022-08-01
> **Response to reviewer da9e**
>
> - The explanation on lines 177-184 is not very clear.
> The last sentence of the paragraph is a little puzzling. Looking at the supplementary material mostly clears this up.
> - There are some typos in the supplementary material: at least one missing reference, one typo that says
> “n < n_0” rather than “n > n_0” while referencing (23). Please proofread this part of the paper.
>
> We will clarify the writing in the main document and correct the typos in the supplementary.
> We will fix the broken references, for example, on line 528 and the caption of table 2
> which should read Algorithms 1a-1c.  The n < n_0 on line 649 will be corrected.
>
> - Do you think it’s possible to prove privacy for these sketches if the hash function is public but the  stream is preprocessed in some random way?
>
> Unfortunately, no.  When the hash function is public there is no randomness in the algorithm except for the dummy items in the sketch initialization. Thus, we'd expect that any preprocessing operation would already need to protect the privacy of all items before passing it to the sketch, which is an even harder task. Alternatively, one could keep a public hash but add private noise to the sketch, as is done in [21, 22].

---

> > ### Comment · Reviewer_da9e · 2022-08-05
> > **Thank you for answering my questions**
> >
> > Thank you for the clarifications!

---

### Author Response · Authors · 2022-08-01
**Thank you for your helpful comments.**

We thank the reviewers for their thoughtful feedback and detailed comments. We're especially thankful that reviewer 4 enjoyed the paper!

---

### Meta-Review · Area_Chair_ofpb · 2022-08-25

**Recommendation:** Accept
**Confidence:** Certain

**Metareview:**

Differentially privacy is often an important design constraint in algorithms, and previous work has studied the question of designing streaming algorithms that satisfy DP in addition to other typical streaming algorithm desiderata (small space, small update time). For the problem of cardinality estimation, previous work has shown that specific streaming algorithms are, or can be modified to be, differentially private. This work shows a general result saying that any streaming algorithm for cardinality estimation, under mild assumptions, is differentially private. This is a clean unified result that the reviewers found appealing. I am in agreement and recommend acceptance.


**Award:**

No

---

### Decision · Program_Chairs · 2022-09-14

Accept